

# Analysis of Soil Hydraulic and Thermal Properties for Land Surface Modelling over the Tibetan Plateau

Hong Zhao, Yijian Zeng, Shaoning Lv, Zhongbo Su

Department of Water Resources, Faculty of Geo-Information Science and Earth Observation (ITC), University of
Twente, Hengelosestraat 99, Enschede 7514AE, The Netherlands.

*Correspondence to*: Hong Zhao (h.zhao@utwente.nl)

**Abstract**

Soil information (e.g. soil texture and porosity) from existing soil datasets over the Tibetan Plateau (TP) is claimed
to be inadequate and even inaccurate for determining soil hydraulic properties(SHP) and soil thermal properties
(STP), hampering the understanding of the land surface process. As the soil varies across three dominant climate
zones (i.e. arid, semi-arid, and semi-humid) over the TP, the associated SHP/STP is expected to vary
correspondingly. To obtain an explicit insight into the soil hydro-thermal consistency for land surface modelling
over the TP, *in situ* and laboratory measurements of over 40 soil property profiles were obtained across the climate
zones. Results show that porosity and SHP/STP differ across the climate zones and strongly depend on soil texture.
In particular, it is proposed that gravel impact on porosity and SHP/STP are both considered in the arid zone and in
deep layers of the semi-arid zone. Parameterization schemes for porosity, SHP and STP are investigated and
compared with measurements taken. This reveals that the porosity determined by the bulk density scheme is the
most applicable for the TP. To determine the SHP, including soil water retention curves and hydraulic
conductivities, the pedotransfer functions (PTFs) developed by Cosby et al. (1984) (for the Clapp-Hornberger
model) and the continuous Wösten et al. (1999) (for the van Genuchten-Mualem model) are recommended. The STP
parameterization scheme proposed by Farouki (1981) based on the mode of De Vries (1963) performed better across
the TP than other schemes. Using the parameterization schemes mentioned above, the uncertainties of five existing
regional and global soil datasets and their derived SHP/STP over the TP are quantified through comparison with *in*
*situ* and laboratory measurements. The measured soil property dataset is available at
http://doi.org/10.4121/uuid:61db65b1-b2aa-4ada-b41e-61ef70e57e4a.

Keywords: Soil hydraulic and thermal properties; the Tibetan Plateau; Pedotransfer functions; Soil Maps; Land
Surface Model.



## 1. Introduction

As the highest plateau in the world with a very large area, the Tibetan Plateau (TP) exerts a huge influence on the Earth's climate system (Yao et al., 2012; Qiu, 2008; Ma et al., 2017; Kang et al., 2010). Studying this influence can advance our understanding of climate change (Ma et al., 2017). Soil Moisture (hereafter as SM) - one of the lower boundary conditions of the atmosphere - is a crucial land surface state (Koster et al., 2004) and therefore of high interest. It modulates land-atmosphere interactions and reflects the trend and the variability of feedback between the water cycle and climate over the TP (Su et al., 2013; Su et al., 2011). Consistent spatial-temporal SM data can be obtained by using land surface models (LSMs) assimilating *in situ* and satellite observations. In these models, the specification of soil hydraulic properties (SHP) (i.e. soil water retention curve, hydraulic conductivities) and soil thermal properties (STP) (i.e. thermal conductivities and heat capacity) is considered more decisive for SM simulation than atmospheric forcing and land surface characteristics are (Shellito et al., 2016; Livneh et al., 2015; Kishné et al., 2017; Gutmann and Small, 2005). SHP and STP govern the partitioning of SM between infiltration and evaporation flux and water heat transport processes (Zeng et al., 2009a; Zeng et al., 2009b).

Basic soil properties such as soil texture and porosity determine the SHP and STP, and show spatial variation over the TP due to the varying formation factors (e.g. climate & parent material). The TP could be categorized into three dominant climatic zones: an arid zone, a semi-arid zone and a semi-humid zone, according to the Food and Agriculture Organisation (FAO) Aridity Index Map (Zeng et al., 2016). The Tibetan Plateau Observatory of plateau scale soil moisture (SM) and soil temperature (ST) (Tibet-Obs) (Su et al., 2011) are distributed throughout these climatic zones: the Ngari network in the arid zone, where sandy soils mixed with gravel are distributed widely; the Naqu network in the semi-arid zone, where loamy sand with organic matter and gravels dominate; the Maqu network in the semi-humid zone, where fine minerals with large silt proportions prevail. Of these, the Naqu network is collocated with the multi-scale SMST monitoring network on the central Tibetan Plateau (CTP-SMTMN) (Yang et al., 2013). *In situ* measurement of soil basic properties and SHP & STP across these networks is crucial for understanding the soil physical consistency of LSMs, and thereby, understanding the land-atmosphere interactions over the TP.

LSMs frequently used are the Clapp and Hornberger (1978) model (i.e. CH scheme) and the Van Genuchten (1980)-Mualem (1976) model (i.e. VG scheme) for SHP, and the Farouki (1981) and Johansen (1975) schemes for STP. Pedotransfer functions (PTFs) (Bouma, 1989) using basic soil property information are developed to estimate parameters in the above SHP/STP schemes. For example, the Noah and Community Land Model (Chen and Dudhia, 2001; Oleson et al., 2008) used the Cosby et al. (1984) PTF for the CH scheme. The Hydrology-Tiled European Centre for Medium-Range Weather Forecasts (ECMWF) Scheme for Surface Exchanges over Land (H-TESSEL) used the soil class PTF (e.g. based on soil texture types) for the VG scheme (Balsamo et al., 2009).

Global and local efforts have been made to compile and develop soil databases, such as the FAO-UNESCO Soil Map of the World (2007) (hereafter as *FAO-UNESCO*), the Harmonized World Soil Database (hereafter as *HWSD*) (FAO/IIASA/ISRIC/ISSCAS/JR, 2012), a Chinese data set of soil properties (Shangguan et al., 2012; Shangguan et al., 2013) and soil hydraulic parameters using PTFs (Dai et al., 2013) released by Beijing Normal University (hereafter as *BNU*), "SoilGrids1km" (Hengl et al., 2014) and the updated version of "SoilGrids250m" (Hengl, 2017) released by the International Soil Reference and Information Center (ISRIC)—World Soil Information (WoSIS) Institute, and the hydraulic parameters based on SoilGrids1km and Schaap et al. (2001) PTFs (hereafter as *HPSS*) (Montzka et al., 2017). The soil profiles collected *in situ* as metadata for developing the above datasets were accessible from ISRIC Word Data Center for Soils (Batjes et al., 2017). Nevertheless, when the metadata were extracted over the TP, only soil texture fraction and soil organic content profiles were available. Profiles of other vital soil properties, such as bulk density (BD) and porosity, were not provided (e.g. no *in situ* BD or porosity profiles available). Furthermore, the basic soil properties from above cited datasets were not necessarily consistent with each other. Uncertainties in soil datasets over the TP might cause bias in predicting SHP & STP, and hence introduce uncertainties in representing the land surface states by LSM. It has been reported that the overestimations of ECMWF SM analyses in the central TP could be partly attributed to the unrepresentative soil information from the FAO Digital Soil Map (2003) as used in H-TESSEL (Su et al., 2013).



Studies using information on state variables (e.g. near-surface soil moisture or brightness temperature) can retrieve effective SHP & STP directly or indirectly through PTFs and LSMs (Ines and Mohanty, 2008a; Han et al., 2014; Dimitrov et al., 2014; Dimitrov et al., 2015; Yang et al., 2016). Nevertheless, most of such retrievals only focus on the basic surface soil properties and SHP, based on the assumption of a homogenous soil column. If the system is highly heterogeneous (e.g. along the vertical profile), retrieval may be problematic (Ines and Mohanty, 2008b).

In this study, based on the *in situ* and laboratory measurements of soil physical property profiles, the variations in basic soil properties and SHP & STP across the three climate zones of the TP are investigated. The parameterization schemes of porosity and SHP & STP are discussed for their applicability over the TP for land surface modelling. Furthermore, the uncertainties of the five existing regional and global soil datasets and their derived SHP & STP over the TP are quantified against the *in situ* and laboratory measurements. In section 2 of this paper, the field campaign and laboratory experiments are introduced as well as the parameterization schemes for porosity and SHP & STP estimates. Results on the soil basic property profiles and SHP & STP profiles over the TP are discussed in Section 3. The evaluation of existing soil datasets is also presented. The availability of the measured soil property dataset is documented in Section 4. Conclusions are presented in Section 5.




## 2. Experimental Designs and Parameterization Schemes

### 2.1 Experimental Designs

#### 2.1.1 Field experiment

A field experiment was carried out across the TP in August 2016, taking soil core samples and measuring field saturated hydraulic conductivity ($K_s$) at various soil depths (Table 1 & Fig. 1). Soils were vertically sampled using sample rings and augers (Eijkelkamp Soil & Water Company) in the vicinity of existing SMST stations of the Tibet-Obs (Su et al. 2011). Table 1 lists the specific sampling approach: 1) the soil was sampled (c.a. 200 g) with a plastic bag used to measure gravel content, soil texture and soil organic content (SOC); 2) the soil was sampled with standard sample rings (5cm in height, 100 cm$^3$ in volume) for the determination of bulk density (BD), porosity and thermal conductivity ($\lambda$); 3) for deriving the Soil Water Retention Curve (SWRC), a dedicated small sample ring (1 cm in height, 20 cm$^3$ in volume) was used; 4) the *in situ* $K_s$ was measured using the Aardvark permeameter (2840 operating instructions - Eijkelkamp), a fully automated constant-head borehole permeameter. The Reynolds and Elrick solution aided with soil texture-structure category information (Elrick, 1989) was chosen for calculating $K_s$.

Within the Maqu network, soil samples were collected at eight stations, located in areas to the east, west and southeast of the ELEBARA-III radiometer location as well as in the southwest corner of the Maqu network (Fig. 1a). The $K_s$ was measured at three locations near the ELBARA station, and at one location (CST05-near) in the southwest corner. Within the Naqu network, soil samples were taken at eight sites along the southwest branch of the CTP-SMTMN network by Yang et al. (2013) (Fig. 1b). The $K_s$ was measured at seven sites at BJ, Naqu_west, NQ01-04 and MS3608. Within the Ngari network, we sampled soils at 14 stations (Fig. 1c). Eight sites at Ali02, SQ03, SQ07, SQ10, SQ17, SQ18, SQ20 and SQ21 were chosen for $K_s$ measurement. In total, 155 soils samples were taken and loaded into plastic bags, 101 samples were collected in standard rings and another 96 samples in small rings. Due to the remoteness and harsh environment on the TP, the locations chosen for soil sampling and fieldwork needed to take practical considerations into account, such as: 1) the location should be accessible by track, local road or national road; 2) the surrounding area should be flat enough to be representative of the local area.

#### 2.1.2 Laboratory Experiment

Three categories of soil samples were handled. From the 155 samples (59 from Ngari, 45 from Naqu and 51 from Maqu) in plastic bags, the soils were first separated into coarse and fine components by using a sieve of 2 mm diameter mesh and weighed separately to obtain gravimetric gravel fractions (GGF). Based on fine minerals and the standard particle size classes of the United States Department of Agriculture (USDA), sand and clay percentages as well as the mean particle diameter of the fine component (FD) were determined using the Malvern Mastersizer 2000 particle size analyzer (http://www.malvern.com), and the SOC was determined by the Total Organic Content analytical instrument of Multi N/C 3100 (http://www.analytik-jena.de/). For the coarse components, a set of sieves with diameters of 2, 2.5, 4, 5, 7, 10, 16, 20, 25, 31.5, 40 and 50 mm were used to obtain their particle size distribution and the mean particle diameter of gravels (GD).

The 101 undisturbed soil samples (35 from Ngari, 21 from Naqu and 45 from Maqu) in standard sample rings were saturated and then dried in the oven (105°C) for 24 hours. The difference between wet and dry weight with known volume was used to calculate porosity and BD. The KD2Pro thermal property analyzer connected to an SH-1 sensor (Decagon Devices) was used to measure heat capacity $C_s$ and thermal conductivity $\lambda$, while the soil was drying, providing drying $C_s$-SM and $\lambda$-SM curves.

The 96 samples in small rings were intended for use in the SWRC experiment using the pressure cell method, but to complete this entire task was considered too time and labor consuming. Therefore, instead of utilizing all soil samples, we regarded30 samples at E-east, E-west, E-southwest, CST05-near, NST30 and NST33 sites in the Maqu network to be representative. Soil samples with loose structures as in the Naqu and Ngari networks were found to easily seep out of the thin rings. Therefore, samples packed in standard rings were used at Naqu_north, SQ17, SQ18 and SQ21 sites for measuring SWRC in the semi-arid and arid regions.

The quality of the measured soil property dataset was evaluated based on quality indicators (e.g. observation date, level of trust, data quality rating and accuracy) as from WoSIS (Ribeiro et al., 2015). The results show that the dataset is of trust level 'C', which means the highest level of the subjective measure inferred from soil expert



knowledge. The entered data (level 'A') have been standardized (level 'B') and harmonized (level 'C'). Furthermore, the values of the measured soil properties and SHP & STP were compared to those in the literature to cross-check if they were within a reasonable range.

The collected basic soil property and SHP & STP datasets over Tibet-Obs were further used to evaluate the existing soil datasets of FAO-UNESCO (FAO/UNESCO, 2007), HWSD (FAO/IIASA/ISRIC/ISSCAS/JR, 2012), BNU (Shangguan et al., 2012; Shangguan et al., 2013), SoilGrid1km (Hengl et al., 2014), SoilGrid250km and HPSS (Montzka et al., 2017) over the TP. The detailed information of the datasets used is listed in Table S1 of the Supplement. All datasets were linearly interpolated to match the measured dataset at specific depths, to ensure the (inter) comparability.

**2.2 Parameterization Schemes**

Many basic soil property dependent schemes have been proposed for porosity estimation. The Cosby et al. (1984) PTF that used sand percentage (hereafter the *Cosby-S scheme*) has been widely used. Porosity can also be inversely related with soil dry bulk density (Hillel, 2003) (hereafter the *BD scheme*). In most cases, these schemes perform well. However, with the existence of SOC, soil porosity and water retention capability both tend to increase. As

gravel content increases, these properties tend to decrease. Chen et al. (2012) parameterized the impact of SOC and gravel content into a porosity estimation scheme (hereafter the *SocVg scheme*). Zhang et al. (2011) proposed a mixing-coefficient model to depict porosity of binary mixture made of a coarse (gravel) and a fine component over a range of gravel content (hereafter the *BM scheme*).

In this study, as Fig. 2 shows, the Cosby-S, BD, SocVg and BM schemes were evaluated for their applicability over

the three climate zones. For the SHP estimate, we selected the Clapp and Hornberger (1978) (hereafter CH) and the Van Genuchten (1980) - Mualem (1976) (hereafter VG) schemes. Based on measured SWRCs, we used the scaling method (Montzka et al., 2017) to obtain the hydraulic parameters and derive the field capacity (FC) and the permanent wilting point (WP) (regarded as the SM at about -33 kPa and -1500 kPa of suction pressure, respectively). Furthermore, the selected PTFs (see Appendix, Table A1) combined with the optimal porosity scheme

were used to estimate the SWRCs-CH and SWRCs-VG. These two estimations (e.g. measured vs. PTFs) were compared, and suitable PTFs with the lowest bias were selected.

Saturated hydraulic conductivity, $K_s$, combined with SWRCs-CH or SWRCs-VG is used to calculate the unsaturated hydraulic conductivity ($K$) and diffusivity ($D$). The PTFs used for SWRCs estimation also had equations (see Appendix, Table A1) for estimating $K_s$, while most PTFs were developed based on fine minerals. To estimate the $K_s$

of a mixture containing gravels, Peck and Watson (1979) used a heat-flow analogy correlating with the $K_s$ of fine minerals and the volumetric gravel fraction (hereafter the *PTFs-VGF scheme*). This PTFs-VGF scheme can be applied to soils with low gravel content (Zhang et al., 2011). It is noted that the PTFs-VGF scheme needs an input ($K_{sat,f}$, see Appendix A.4) from the PTFs $K_s$ estimation. Furthermore, Koltermann (1995) used the Kozeny–Carman equation to estimate the hydraulic conductivity for binary mixtures, and a suitable grain diameter estimation was

also declared important (Kamann et al., 2007). To improve the performance of the Kozeny–Carman equation for estimating the $K_s$ of binary mixtures, Zhang et al. (2011) introduced the BM scheme for estimating porosity and a power-averaging method for calculating representative grain diameter (hereafter the *BM-KC scheme*). In this study, the PTFs (see Appendix, Table A1), PTFs-VGF and BM-KC schemes were employed as shown in Fig. 2.

Several (semi-) empirical models have been developed to estimate the STP. De Vries (1963) developed a Maxwell

equation analogous physics based model to describe $\lambda$. This model can predict $\lambda$ accurately, although this is complicated by the fact that at least five mineral components and their separate shape features need to be taken into account (Tarnawski and Wagner, 1992). Furthermore, the effect of vapor movement caused by the temperature gradient is also considered in the De Vries (1963) model. It should be noted that the consideration of soil vapor flow is critical to accurately investigate the simultaneous transfer of moisture and heat, particularly in semi-arid and arid

environments (Zeng et al., 2011a, b; Zeng and Su, 2013). Farouki (1981) regarded liquid water as the continuous medium and soil minerals as uniform particles in the De Vries (1963) model. Furthermore, the $\lambda$ of minerals was estimated by using a geometric mean equation from the quartz content in soil minerals and the $\lambda$ of quartz and other minerals. The $\lambda$ of vapor together with the shape feature for air was calculated in terms of water content and porosity



(hereafter the *D63F scheme*). Tian et al. (2016) developed a simple and generalized De Vries-based model, which assumed that the $\lambda$ and shape features of soil minerals were determined by soil texture (sand, clay and silt), and that the effect of vapor movement was negligible (hereafter the *T16 scheme*). The empirical model proposed by Johansen (1975) used the Kersten (1949) number and $\lambda$ in dry and saturated conditions to estimate $\lambda$ (hereafter the *J75 scheme* ). In this study, as shown in Fig. 2, the D63F, T16 and J75 schemes were adopted. For each $\lambda$ scheme, a comparison was made using parameters (i.e. the $\lambda$ of minerals) with and without gravel and SOC considerations.

The De Vries (1963) model was used for calculating $C_s$. The details of porosity and SHP & STP schemes are listed in the Appendix (A.1–5).



### 3. Results and Discussions

**3.1 Basic soil properties (*in situ*)**

*Soil texture*

Figure 3 shows the mean of sand, clay and silt percentages, gravimetric gravel fraction (GGF), soil organic content (SOC), and the diameter of fine components and gravels (FD and GD) at profiles across the three climate zones over the TP. In the arid zone (Fig. 3a), the sand content was around 80% and exhibited higher values at surface layers of 5 and 10 cm than at deep layers. Silt and gravel contents ranged between 10-20% and this percentage increased with 205 depth. Clay content and SOC were 3% and 0.8%, respectively, and remained constant along the profile. The FD and GD ranged from 0.19-0.24 mm and 4-8 mm, respectively, and showed a tendency to increase from the top to a depth of 20 cm, but to decrease in the deeper layers. It can be concluded that soil texture in the arid zone consisted of a high proportion of coarse sand accompanied by gravel, and that the gravel content increased the deeper the layer.

In the semi-arid zone (Fig. 3b), the mean sand fraction ranged from 70-80% with a slight decrease with depth. The 210 silt and clay contents ranged from 15-25% and 4-8%, respectively, and displayed an increasing trend with depth. The GGF at depths of 40 and 50 cm exceeded 50%, while it was much lower for the shallow layers. Mean FD and GD ranged from 0.18-0.22 mm and 4-8 mm, respectively. GD at deep layers was larger than that at shallow layers. SOC approached 10% in the surface layers but quickly declined at deep layers. It can be summed up that soil texture in the semi-arid zone was dominated by a high percentage of sand mixed with a small proportion of gravels, but with 215 high SOC in shallow layers, and mainly mixed with big gravels at deep layers.

In the semi-humid zone (Fig. 3c), mean silt and clay contents were around 60% and 10%, respectively, with a smoothly decreasing trend along the profile. The mean sand fraction ranged from 28-40% and increased with depth. No gravel was found. Mean FD ranged from 0.024-0.036 mm, and fine particles in deep layers (40 and 80 cm) were larger than in shallow layers (see Fig. 3c lower panel). Similar to the SOC profile distribution in the Naqu network, 220 stratification occurred with SOC in surface layers almost reaching 20%, and declining at almost double the rate at each further depth. Soil texture in the semi-humid zone was characterized as being dominated by a high percentage of silt content with relatively large SOC in the shallow layers, and with mainly fine sand mixed in the deep layers.

*Bulk density & Porosity*

In the arid zone (Fig. 4a), the BD varied slightly (between 1.55 and 1.65 g/cm$^3$) with depth, showing a peak at 10cm. 225 The porosity of the surface layer was slightly higher than in deep layers, with a mean profile porosity of 0.33. The porosity at 20 cm was the lowest in the profile, which might be caused by this layer containing the greatest proportion of gravel as well as the greatest GD and FD (see Fig. 3a, upper and lower panels). In the semi-arid zone (Fig. 4b), the BD increased continuously with depth, with a minimum of 1 g/cm$^3$ in the top layer and a maximum of 2.1 g/cm$^3$ in the bottom layer. The porosity peaked at around 0.6 in the top layer, while monotonously decreasing to 230 0.25 down into the bottom layer. Combined with the soil texture analysis (see Fig. 3b), variations of BD and porosity in the profile were inferred relevant to the high SOC in the surface layer and the large gravel content in the bottom layer. In the semi-humid zone (Fig. 4c), BD ranged from 0.8 to 1.5 g/cm$^3$ and increased with depth, while porosity decreased with depth and ranged from 0.72 to 0.45. The stratification of BD and porosity in the profile might be induced by SOC laying as Fig. 3c reveals. In summary, profiles of BD and porosity differ with soil texture 235 variation over the three climate zones, and both the SOC and gravels affect the porosity. Overall porosity at shallow layers (5, 10 and 20 cm) decreased from the arid to the semi-arid and to the semi-humid zones, while at deep layers (>= 40cm) it showed a decrease from the semi-arid to the arid and then the semi-humid zones.

*Saturated hydraulic conductivity ($K_s$)*

In the arid zone (Fig. 5a), the magnitude of mean $K_s$ was 10$^{-5}$ (m/s). The $K_s$ at a depth of 20 cm was lower than at 240 other depths, which might be due to this layer exhibiting least porosity (see Fig. 4a). In the semi-arid zone (Fig. 5b), the mean $K_s$ exhibited two orders of magnitude with depth, namely 10$^{-6}$ (m/s) at depths of 10, 20 and 50 cm and 10$^{-5}$ (m/s) at a depth of 40 cm. In the semi-humid zone (Fig. 5c), $K_s$ also differed two orders of magnitude: 10$^{-6}$ (m/s) at depths of 5, 10, 20 and 80 cm and 10$^{-7}$ (m/s) at a depth of 40 cm. It is to be noted that the $K_s$ profiles of both the semi-arid and semi-humid zones depict a lower $K_s$ in shallow layers than in the deeper layer. This is mainly due to





the negative correlation between saturated hydraulic conductivity and soil organic carbon in soil (Nemes et al., 2005; Wang et al., 2009). As can be seen, $K_s$ varied with soil texture over the three climatic zones, and both SOC and gravel had an effect. At a certain depth, where the soil basic properties underwent a transition (see Fig. 3), a minimum $K_s$ always existed. The mean and the standard deviation of the soil properties of the profiles in the three climate zones are listed in the Supplement (Table S2-S4).

*Gravel impact on porosity and $K_s$*

Figures 6a&b show that with a GGF < 0.3 in shallow layers, porosity did not tend to change, while with a GGF > 0.4, porosity tended to decline, especially in deep layers. For example, porosities with a GGF of 0.6 and 0.72 at 20 cm and 40 cm depths were lower than those with a GGF < 0.3 at 5 cm and 10 cm depths (Fig. 6a). With more gravels embedded in the matrix, the flow paths in the soil would become blocked and the porosity reduced (Zhang et
al., 2011). However, the porosity did not always decrease as the GGF increased. Porosity with a GGF of 0.84 in the semi-arid zone was higher than porosities with a GGF ranging between 0.4 and 0.6 at 50 cm depth (Fig. 6b). Porosity with a GGF of 0.7 at 20 cm depth in the arid zone was also higher than that with a GGF of 0.6 at 40 cm depth (Fig. 6a). Porosity tended to increase as the GGF increased. In fact, when a GGF is relatively high (> 1-porosity of gravels), connected pores can form among the gravels, and thus increase porosity (Zhang et al., 2011).

Figures 6c&d show a slight decrease in $K_s$ at 10 cm depth at a GGF < 0.62 and a slight increase in $K_s$ at depths of 20 and 40 with a GGF > 0.8, which is consistent with the changes in porosity. The observations clearly show that gravels have a distinct impact on the porosity and $K_s$ in the arid and semi-arid zones. It should be noted that, although the *in situ* $K_s$ measurements were conducted at locations adjacent to the places where we took soil samples, the heterogeneity issue may still exist. Nevertheless, the current findings based on field experiments are in line with
reported findings based on laboratory experiments (Zhang et al., 2011; Koltermann, 1995; Sakaki and Smits, 2015).

*Heat capacity $C_s$ and thermal conductivity λ*

Figures 7a&b&c show that no distinct stratification existed for $C_s$ profiles with SM varying over the three climate zones. The $C_s$ ranged from 1-2.5 MJ m$^{-3}$ K$^{-1}$ over the arid zone, 0.5-3 MJ m$^{-3}$ K$^{-1}$ over the semi-arid zone and 0.5-2.4
MJ m$^{-3}$ K$^{-1}$ over the semi-humid zone as soils dried out.

Figures 7d&e&f show the relationship of λ-SM varied with depth and depended on soil texture. For the arid zone (Fig. 7d), the λ-SM curves are very similar at each depth due to the nearly homogenous sandy soils across the whole profile (see Fig. 3a). The mean λ ranged from 1.8-0.2 (W m$^{-1}$ K$^{-1}$) as the soils dried out. In the semi-arid zone (Fig. 7e), the λ-SM curves were stratified, and soils with gravels in deep layers (see Fig. 3b) clearly had a higher λ (>2 W
m$^{-1}$ K$^{-1}$) than in other layers and other climate zones. In the semi-humid zone (Fig. 7f), the λ-SM curves also presented layering, though within a much narrower range than in the semi-arid zone. Such layering is mainly caused by the sand distribution along the profile, which increased slightly with depth (see Fig. 3c). The mean λ in the semi-humid zone ranged from 0.2-1.6 (W m$^{-1}$ K$^{-1}$) as soils dried out. Furthermore, the surface layers in the semi-arid and semi-humid zones comprised a lower λ (Figs. 7e&f) because of the SOC influence.

**3.2 Porosity estimation**
Four schemes were used to estimate porosity, and the results were compared with the measured values. In the arid zone, most porosities estimated by the Cosby-S scheme were higher than the measured porosities, and varied across a smaller range (0.36-0.42) than the measurements (0.28-0.42) did (Fig. 8a-1). Nevertheless, the Cosby-S scheme simulated the soil porosity at the surface layer well (smaller bias and RMSE; see Appendix, Table A2). The porosity
derived from the BD scheme is seen distributed along the 1:1 line with the measurements (Fig. 8a-2) with lower bias and lower RMSE (see Table A2). The SocVg scheme strongly overestimated porosities (Fig. 8a-3), while porosities derived from the BM scheme matched well with the measurements (Fig. 8a-4) and comprised the lowest RMSE (see Table A2). As can be seen, for the arid zone, the Cosby-S scheme did not capture the variation in porosity at all depths, because the sand fraction only differed slightly with depth. The BD scheme performed well estimating the
porosity, as it is a bulk estimation scheme that takes both gravel and fine minerals into consideration. Although the



SocVg scheme considers gravel impact through volumetric SOC determination, it is assumed that the gravel has the same porosity as the minerals (e.g. as determined by the Cosby-S scheme). In the SocVg scheme, the porosity of gravels with a theoretical minimum value (0.363) is higher than the maximum (0.31) found by Wu and Wang (2006). Consequently, the SocVg scheme led to the overestimation of porosity here. Because it considers the degree of mixture between fine minerals and gravels, the BM scheme also performed well, especially in the deep layers.

In the semi-arid zone, porosities estimated by the Cosby-S scheme were approximately constant (0.40) (see Fig. 8b-1), which is contrary to the measured porosities (see Fig. 4b). Porosities derived from the BD scheme agreed well with the measurements (see Fig. 8b-2) and had the lowest bias and RMSE at all depths (see Table A2). The SocVg scheme overestimated porosities (see Fig. 8b-3). The BM scheme underestimated porosities at depths of 5, 10 and 20 cm (Fig. 8b-4), but simulated them well at deep 40 and 50 cm layers (lower bias and RMSE in Table A2). In the semi-humid zone, the Cosby-S scheme failed to estimate porosities (see Fig. 8c-1). As in other climate zones, the BD scheme represented the measurements in the semi-humid zone well (see Fig. 8c-2) with low bias and low RMSE (see Table A8), indicating its predictive accuracy and applicability for all depths and all three climate zones. It should be noted that the SocVg scheme performed better at surface layers than at greater depth (see Fig. 8c-3). Taking all these results together, it indicates that the BD scheme delivered the greatest predictive accuracy for estimating porosity in profiles across the three climate zones.

### 3.3 Soil hydraulic properties
*Pressure-cell-determined SWRCs*

Figure 9 shows that the SWRCs differed across the three climate zones and strongly depended on soil texture. Consistent with soil stratification, the SWRCs also showed distinct layering. The CH and VG models captured the retention characteristic of soil water well across the three climate zones. Particularly for the arid and semi-arid regions, both the CH and VG models performed well, with just a slight underestimation by the VG model. In the semi-humid area, the VG model was more consistent with measurements than the CH model for soil suction <300 kPa. On the other hand, the CH model was more accurate when suction increased, although both the CH and VG models overestimated SM at a suction > 300 kPa. Estimated parameters of the CH and VG models are listed in Table .

Various PTFs (see Table A1) combined with the BD porosity scheme were used to predict SM at the different suction levels (i.e. SWRCs-CH & SWRCs-VG) as adopted by the pressure cell experiment. The mean estimated SWRCs from PTFs along with their variation ranges in each profile were compared with the measurements shown in Fig. 10, and their absolute biases are shown in Fig. 11.

Fig. 10a shows the Saxton et al. (1986) PTFs overestimated the SWRCs-CH in the profiles in the arid zone, while the PTFs given by Campbell and Shiozawa (1992) and Saxton and Rawls (2006) underestimated them, and the Cosby et al. (1984) PTFs (1&2) presented good predictions with smaller absolute biases (Fig. 11a). In the semi-arid zone (Fig. 10b), all PTFs underestimated the SWRCs-CH at surface layers, while the Cosby et al. (1984) PTFs (1&2) and Saxton et al. (1986) PTFs captured these well at deep layers and had lower biases in the profile (Fig. 11b). In the semi-humid zone (Fig. 10c), the Saxton et al. (1986) PTFs underestimated SWRCs-CH in the profile, while other PTFs predicted them well, especially the Cosby et al. (1984) PTFs (1&2) (i.e. both had lower biases, see Fig. 11c, with the PTFs (1) performing better at high suction). Therefore, the Cosby et al. (1984) PTFs (1) are considered most suitable for predicting SWRCs-CH over the three climate zones. It is noteworthy that in combination with the BD scheme, the Cosby PTFs (1) performed much better regarding the estimation of SWRC-CH, compared with the estimates by the Cosby PTFs (1) combined with the Cosby-S porosity scheme (see section 3.2). On the other hand, without the BD scheme, the Saxton and Rawls (2006) PTFs were found to be performing better over the semi-arid and semi-humid zones (see Supplement Fig. S1-S2).

For the SWRCs-VG estimates, the Rosetta1-H3 and Rosetta3-H3 PTFs were developed based on the mixed database (Schaap et al., 2001). Fig. 10 (right panel) shows they underestimated SWRCs-VG in the profiles across the three climate zones, as did the Rawls and Brakensiek (1985) PTFs. The Rosetta3-H3 PTFs also underestimated the SWRCs-VG in the semi-arid zone (Fig. 10b), as did the Class Wösten et al. (1999) PTFs. The Vereecken et al. (1989) PTFs, which were developed based on a database where hydraulic properties were measured for every



sample with the same measurement techniques (Vereecken et al., 2010), performed well when the $m$ was set at 1.
However, these PTFs were not performing well for $m=1-1/n$ in the VG model and overestimated heavily in the semi-humid zone (broken green line in Fig. 10c). The Continuous Wösten et al. (1999) PTFs were derived from the database of Hydraulic Properties of European Soils (HYPRES) and as such were more affiliated with the database of Vereecken et al. (1989). The Weynants et al. (2009) PTFs were also developed based on the Vereecken et al. (1989) database and included BD as the variable. These two PTFs predicted SWRCs-VG well for the three climate zones,
with smaller biases, and the former performing better for the semi-arid zone (Fig. 10 & Fig. 11). Consequently, the Cosby et al. (1984) PTFs (1) and the Continuous Wösten et al. (1999) PTFs combined with the BD porosity scheme have demonstrated to be most applicable for predicting the SWRCs-CH and SWRCs-VG, respectively, across the three climate zones.

### $K_s$ estimation

The $K_s$ estimates obtained with the PTFs scheme (see Table A1), the empirical PTFs-VGF scheme (PTFs scheme scaled by gravel content) and the semi-physical BM-KC scheme (see Appendix, A.4) were compared against the *in situ* observations. Figure 12a&b show the PTFs scheme had a lower bias for $\mathrm{Log}_{10}\,K_s$ than the PTFs-VGF and BM-KC schemes for the arid zone. In particular, the PTFs given by Cosby et al. (1984) (1&2) predicted $K_s$ well for the CH model, as did the Rosetta1-H3 PTFs, Rosetta3-H3 PTFs and Rawls and Brakensiek (1985) for the VG model.
The BM-KC scheme had lower RMSE at 40 cm depth, indicating the gravel impact on $K_s$.

Figure 12c&d show that the BM-KC scheme had a lower bias for $\mathrm{Log}_{10}\,K_s$ than most PTFs and PTFs-VGF at depths of 10, 20 and 40 cm in the semi-arid zone. For $K_s$ estimation in the CH model, the Cosby et al. (1984) (1) PTFs performed best at shallow depths, while the PTFs-VGF of these PTFs were better at deep layers of 40 and 50 cm. For $K_s$ estimation in the VG model, no distinct difference existed between PTFs and PTFs-VGF schemes, indicating
that the estimated $K_s$ for the VG model is less affected by gravels. The Rosetta1-H3 PTFs predicted $K_s$ better, with only slight overestimation. Figure 12e&f show most of the PTFs underestimated $K_s$, while the selected PTFs (i.e. CosbY (1) & Rosetta1-H3) in the arid zone also performed relatively well in the semi-humid zone. To sum up, the PTFs resulting from Cosby et al. (1984) (1) and Rosetta1-H3 PTFs are appropriate for the estimation of $K_s$ for the CH and VG models, respectively, across the three climate zones. PTFs-VGF of the Saxton and Rawls (2006)
scheme should be applied in deep layers in the semi-arid zone, where gravel is abundant in the soil.

### 3.4 Heat Capacity and soil thermal conductivities
In Figure 13a the $C_s$ estimate by the De Vries (1963) model depicts a small bias ($< 0.1$ MJ m$^{-3}$ K$^{-1}$) across the arid zone and in shallow layers across the semi-humid zone, but large bias across the semi-arid zone and in deep layers across the semi-humid zone. Considering the impact of SOC in the semi-arid and semi-humid zones, Fig. 13b
indicates an improved $C_s$ estimate for soils at top layers, but a worsening at other depths. Nevertheless, the original formulation of the De Vries (1963) model is used for estimating profile $C_s$ (see Appendix A.5).

The D63F, T16 and J75 schemes combined with the BD porosity scheme were used to estimate the $\lambda$. For the arid and semi-arid regions, the estimation of $\lambda$ considered two scenarios: with (Case 1) and without (Case 2) gravel impact. For the semi-humid region, both with (Case 1) and without (Case 2) SOC impact were considered. Figure 14
shows that the D63F model had a lower bias than other schemes in both cases across the three climate zones, indicating a greater ability to predict $\lambda$. The T16 scheme overestimated $\lambda$, which may be due to its ideal assumption that the $\lambda$ of soil minerals is totally determined by sand, clay and silt particles. The J75 scheme generally underestimated the $\lambda$.

Figure 14 also shows that the D63F scheme improved the $\lambda$ estimate at surface layers in the arid zone and at a depth
of 50 cm when incorporating gravel impact parameterization (lower biases in Case 2). The improvement also occurred with the T16 scheme, while biases tended to be greater for the J75 scheme. In the semi-humid zone biases also became larger for all schemes when SOC impact parameterization was involved. Although parameterization of the SOC impact was demonstrated to improve the $\lambda$ estimate in the top layer (SOC> 12%) over the Eastern TP (Chen et al., 2012; Zheng et al., 2015), it should be noted that the Cosby-S scheme was used instead of the BD
scheme, in these studies for porosity estimate, as adopted in this paper (see Sec. 3.2). Comparison of the $\lambda$ estimate using these two porosity schemes for the semi-humid zone show that the D63F scheme combined with the BD





porosity scheme can predict $\lambda$ well across the three climate zones. When the D63F scheme is combined with the Cosby-S scheme, it also performs well (see Supplement Fig. S3).

### 3.5 Evaluation of the existing soil datasets

The current existing global and regional soil datasets, including FAO-UNESCO (FAO/UNESCO, 2007), HWSD (FAO/IIASA/ISRIC/ISSCAS/JR, 2012), BNU (Shangguan et al., 2012; Shangguan et al., 2013), SoilGrid1km (Hengl et al., 2014), SoilGrid250km and HPSS (Montzka et al., 2017), were extracted for the TP and compared with the *in situ* and laboratory measurements collected in the field. The measured basic soil property dataset was named Tibet-Obs.

*Basic soil property*

Figure 15 shows that all datasets underestimated both the sand fraction and BD in the arid and semi-arid regions, while overestimated them in the semi-humid region. For the silt fraction, the pattern was reversed. Almost all datasets overestimated the silt fraction in the arid and semi-arid regions (only FAO-UNESCO underestimated silt very slightly in the semi-arid region), and underestimated the silt fraction in the semi-humid region. All datasets

overestimated the clay fraction throughout the three climate zones.

The estimates of SOC from all the datasets were within 1% range of the measurements across the arid and semi-arid zones, and within 10% across the semi-humid zone, apart from the FAO-UNESCO data, which underestimated the SOC heavily in this region. Most of the GGF estimates for the arid zone were within 10%, with the FAO-UNESCO data underestimating by 20%. For the semi-arid and semi-humid regions, all datasets consistently, respectively,

underestimated and overestimated the GGF.

The BD scheme was used to derive porosity from the existing datasets. Figure 16a shows that the estimations of porosity were higher than the *in situ* measurement for the arid zone, with the SoilGrid1km and HWSD providing the closest approximations. In the semi-arid zone (Fig. 16b), all datasets underestimated porosity at the top layer, but overestimated it at other depths. It should be noted that SoilGrid1km and SoilGrid250m presented porosity almost as

a constant figure in each profile, which is not representative for conditions in the field. The porosity estimations from FAO-UNESCO, HWSD and BNU did show profile variation, although much less than the *in situ* measurements did. In the semi-humid region (Fig. 16c), all datasets underestimated porosity in the surface layers 5, 10 and 20 cm, and either underestimated or overestimated porosity in the deep layers.

*SWRC*

As previous analysis of PTFs (see section 3.3) suggested, the Cosby et al. (1984) and continuous Wösten et al. (1999) PTFs were used to estimate, respectively, SWRCs-CH and SWRCs-VG models. Given the relatively homogenous soil profile derived from existing products (see Fig. 16), the averaged SWRC over different depths along with its deviations in the profile were used for comparison with the laboratory measurements. Figure 17a shows all datasets overestimated SWRCs in the profile in the arid zone, in the order of FAO-UNESCO > BNU >

HWSD > SoilGrid250m > (HPSS for VG model) > SoilGrid1km > Tibet-Obs. In the semi-arid zone (Fig. 17b), all datasets underestimated SWRCs at the surface layers of 5 and 10 cm, while overestimated at deep layers. FAO-UNESCO captured the SWRCs-CH at surface layers well, and BNU presented the closest estimations for deep layers. Regarding SWRCs-VG, SoilGrid250m and HWSD, respectively, matched the measurements at surface and deep layers well. In the semi-humid zone (Fig. 17c), all datasets showed similar SWRCs-CH, slightly

underestimating at low suction (< 100 kPa) but then becoming consistent with the measurements. The results for SWRCs-VG were quite diverse. The HWSD and HPSS showed consistent underestimation. The FAO-UNESCO and BNU closely matched the measurements in deep layers. The SoilGrid1km and SoilGrid250m were within the range of the measurements across the whole profile, although their mean values were larger at high suction range (>300 kPa). Furthermore, it should be noted that the averaged profile SWRCs derived from Tibet-Obs tended to reflect

SWRCs at deep layers over the three climate zones.

Figure 18 shows all existing datasets overestimated FC and PWP in the arid and semi-arid zones, while underestimated them in the semi-humid zone, in comparison with the laboratory measurement. With the Tibet-Obs




dataset as input to the applicable PTFs good FC and PWP were generated. The SoilGrid1km-, HWSD- and SoilGrid1km-derived FC and PWP were close to the mean measured values over the three respective climate zones.


### $K_s$

Figure 19 shows predicted $K_s$ ($10^{-6}$ m/s) in the profile for all existing datasets across the three climate zones. They were of a smaller order than the field measurements in the arid and semi-arid zones but of a larger order than some of the field measurements in the semi-humid zone. The Tibet-Obs dataset as input in the applicable PTFs predicted

$K_s$ well. The existing datasets for estimating SWRCs: SoilGrid1km, HWSD, and SoilGrid1km, also performed well estimating $K_s$ in the three climate zones, respectively.

### *Soil diffusivity (D) and conductivity (K)*

When the soil desaturates, and the largest pores in the soil drain, the hydraulic conductivity ($K$) and diffusivity ($D$) are reduced many orders of magnitude from saturation to dryness (Bittelli et al., 2015). Lower $K$ combined with

higher $D$ will lead to slower water transport, and thereby subsequently a higher SM derived from the land surface model, and vice versa.

Figure 20 and Figure 21 show how $K$ and $D$ vary within climate zones. All datasets provide lower $D$ and $K$ derived from CH and VG models than the laboratory measurements did in the arid zone (Fig20a and Fig21a). Similar to in the SWRCs comparisons, SoilGrid1km and HPSS provided the closest approximation to those datasets in the order:

SoilGrid250m > HWSD & BNU > FAO-UNESCO (Fig20a and Fig21a). In the semi-arid zone, all datasets generated CH-$D$ and CH-$K$ similar to those of the measurements at depths from 20 to 50 cm, while overestimating them at the surface layer (Fig. 20b). All datasets predicted VG-$D$ and VG-$K$ well at depths from 10 to 40 cm, while underestimating them for the deepest layer (Fig21b). A special case is formed by the FAO-UNESCO dataset, which slightly overestimated VG-$K$ at surface layers and heavily underestimated it at deep layers, while heavily

overestimating VG-$D$ at surface layers and slightly underestimating it at deep layers. This would lead to the overestimation of derived SM, as the ECMWF SM analyses do in this region (Su et al., 2013). In the semi-humid zone, all datasets generated high CH-$D$ and CH-$K$, and were close to the measurements at the greatest depth (Fig. 20c). The HPSS overestimated VG-$D$ and VG-$K$. The SoilGrid1km and SoilGrid250m both underestimated VG-$K$, while they could reproduce the VG-$D$ very well. The FAO-UNESCO, HWSD and BNU derived VG-$K$ closely

matched the measurements in the profile, while they slightly overestimated the VG-$D$.

### *Heat capacity $C_s$ and thermal conductivity $\lambda$*

The soil heat transport process is mainly characterized by the soil heat capacity ($C_s$) and thermal conductivity ($\lambda$) (Hillel, 2003). Lower thermal conductivity with higher heat capacity will lead to slower heat transport, and thereby subsequently the higher soil temperature derived from the land surface model, and vice-versa.

Figure 22a shows all datasets underestimated $C_s$-SM and $\lambda$-SM in the arid zone. The degree of undervaluation decreases in the following order: BNU & FAO-UNESCO > SoilGrid250m > HWSD > SoilGrid1km > Tibet-Obs. In the semi-arid zone (Fig. 22b), the HWSD predicted a $C_s$-SM close to the measurements for the top layer. The other datasets underestimated $C_s$-SM when soils became dry, while aligning well with the measurements when the soil became wet (SM>3 cm$^{-3}$ cm$^{-3}$). A special case is formed by the FAO-UNESCO dataset, which underestimated $C_s$-

SM at surface layers, but overestimated $\lambda$-SM, while at other depths estimating $C_s$-SM well, but underestimating $\lambda$-SM. These results would lead to the underestimation of the derived soil temperature, and a soil temperature in a surface layer lower than in a deep layer, which is consistent with the findings of ECMWF soil temperature analyses (Su et al., 2013). All datasets generated $\lambda$-SM close to the measurements at surface layers, with HWSD performing the best when soil became wet. In the semi-humid region (Fig. 22c), all datasets overestimated $C_s$-SM, with the

SoilGrid1km estimation matching the measurements best. All datasets overestimated $\lambda$-SM at surface layers, while underestimating $\lambda$-SM at deep layers, with SoilGrid1km derived $\lambda$-SM being consistent with the mean of the Tibet-Obs dataset.



## 4. Data availability

The soil physical dataset is available at the 4TU.ResearchData data center at http://doi.org/10.4121/uuid:61db65b1-
b2aa-4ada-b41e-61ef70e57e4a. The data is stored in .XLSX files. A readme file describes the structure of the
EXCEL, the measurement devices and contact information. The download linkages of existing soil property datasets
used in this paper are included in .txt file. The location of sampling is stored in .kmz file.

## 5. Conclusions

For this study an *in situ* measurement dataset of soil physical properties was set up across the arid, semi-arid and
semi-humid climate zones across the Tibetan Plateau. Analyzing this *in situ* dataset has made clear how soil texture,
bulk density (BD), porosity, soil hydraulic properties (SHP, i.e. soil water retention curve, hydraulic conductivity)
and thermal properties (STP, i.e. heat capacity and thermal conductivity) differ for each climate zone and vary
within each profile. Soil physical properties present stratification in the semi-arid and semi-humid zones. Gravels
were found affecting porosity and SHP & STP in the arid zone and in deep layers of the semi-arid zone.

Various schemes for estimating the porosity and SHP & STP were examined. The BD scheme has been
demonstrated to be most suitable for estimating the porosity over the three climate zones. The Cosby et al. (1984)
PTFs proved more applicable for SHP estimation by the Clapp and Hornberger (1978) (CH) model, and the
continuous Wösten et al. (1999) PTFs for SHP estimation by the Van Genuchten (1980) - Mualem (1976) (VG)
model. The original formulation of the De Vries (1963) model can be deployed for estimating the heat capacity of a
profile. Furthermore, the De Vries (1963) model combined with the Farouki (1981) scheme (D63F) and with the
implementation of the BD porosity scheme proved superior for estimating thermal conductivity.

Referenced by the measurements, uncertainties of the existing soil basic property datasets and their derived SHP &
STP were quantified across the TP. This information provided indicative signs for soil parameterization in LSMs.
On the other hand, the existing soil property datasets were also used as the ancillary data for SM retrieval. For
example, the composited datasets of FAO and HWSD were used in the Soil Moisture and Ocean Salinity (SMOS)
and Soil Moisture Active Passive (SMAP) SM product generation. Therefore the information also became valuable
for understanding uncertainties in SM products inherited from soil maps. Based on the dataset comparison, this
paper indicates that SoilGrid1km can reduce such uncertainty and is therefore recommended for use in the arid and
semi-humid zones, while the combination of FAO-UNESCO at shallow layers and HWSD at deep layers is
recommended for the semi-arid zone over the TP.

In summary, this paper presents the applicable schemes to use for porosity and SHP & STP estimation in the LSM
across the TP, which is significant for consistent spatial-temporally soil moisture simulation and helps provide the
references for soil basic properties and SHP/STP retrieval. Furthermore, the evaluation of the existing soil property
datasets is crucial for quantifying the uncertainty arising from soil data used in the LSM and in soil moisture
retrieval from microwave remote sensing.

**Author contributions:** Yijian Zeng and Zhongbo Su conceptualized the experiment design; Hong Zhao designed
the experiment with the help of Yijian Zeng; Hong Zhao conducted the field experiment and lab experiment,
summarizing/organizing data; the authors, including Shaoning Lv, were involved in the fieldwork; Hong Zhao wrote
the paper. Yijian Zeng and Zhongbo Su commented and revised the paper.

**Acknowledgements:** We acknowledge the financial support from the Chinese Scholarship Council (CSC) for Hong
Zhao. We thank Prof. Yaowen Xie and Dr. Jizong Jiao for their assistance with the laboratory experiment in
Lanzhou University, China, and Dr. Xin Wang, Dr. Hui Tian, Mr. Zuoliang Wang and Jinlei Chen for their help
with field soil sampling. We also appreciate Prof. Montzka providing the code for the parameter fitting algorithm
and Mrs. Eva Skidmore doing English editing.

**Conflicts of Interest**: The authors declare no conflict of interest.



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





**Tables**

**Table 1 Sampling approach for soil basic properties, SHP and STP over the Tibet-Obs**

| Sampling Approach | Soil Texture | Gravel Content | Soil Organic Content | Bulk Density | Porosity | Thermal Conductivity | Soil Water Retention Curve | Hydraulic Conductivity | Sampling Depths | | |
|---|---|---|---|---|---|---|---|---|---|---|---|
| | | | | | | | | | Maqu | Naqu | Ngari |
| Plastic bag | √ | √ | √ | | | | | | 5cm | 5cm | 5cm |
| Standard sample rings | | | | √ | √ | √ | | | 10cm 20cm | 10cm 20cm | 10cm 20cm |
| Small sample rings | | | | | | | √ | | 40cm 80cm | 40cm 50cm | 40cm |
| Profile Auger | | | | | | | | √ | 10cm 20cm 40cm 80cm | 10cm 20cm 40cm 50cm | 10cm 20cm 40cm |

**Table 2 Pressure-cell determined parameters of the CH and VG models for the three climate zones.**

| Region | Depth (cm) | CH | | | | | | VG | | | | |
|---|---|---|---|---|---|---|---|---|---|---|---|---|
| | | $\lambda$ | $\psi$ | $\theta_s$ | $\theta_{fc}$ | $\theta_{wp}$ | $\theta_r$ | $\theta_s$ | $\alpha$ | n | $\theta_{fc}$ | $\theta_{wp}$ |
| | | - | cm | cm³ cm⁻³ | cm³ cm⁻³ | cm³ cm⁻³ | cm³ cm⁻³ | cm³ cm⁻³ | cm⁻¹ | - | cm³ cm⁻³ | cm³ cm⁻³ |
| Ngari (arid) | 5 | 0.185 | 4.346 | 0.304 | 0.209 | 0.103 | 0.028 | 0.250 | 0.021 | 1.393 | 0.223 | 0.086 |
| | 10 | 0.162 | 5.825 | 0.323 | 0.244 | 0.131 | 0.028 | 0.285 | 0.022 | 1.305 | 0.256 | 0.116 |
| | 20 | 0.160 | 2.023 | 0.270 | 0.173 | 0.094 | 0.028 | 0.201 | 0.020 | 1.340 | 0.182 | 0.082 |
| | 40 | 0.180 | 2.447 | 0.277 | 0.173 | 0.087 | 0.028 | 0.211 | 0.028 | 1.366 | 0.182 | 0.075 |
| Naqu (semi-arid) | 5 | 0.071 | 0.020 | 0.506 | 0.299 | 0.228 | 0.027 | 0.328 | 0.050 | 1.097 | 0.302 | 0.225 |
| | 10 | 0.100 | 11.211 | 0.433 | 0.389 | 0.265 | 0.037 | 0.439 | 0.039 | 1.151 | 0.397 | 0.254 |
| | 20 | 0.162 | 4.594 | 0.393 | 0.286 | 0.154 | 0.037 | 0.350 | 0.036 | 1.285 | 0.299 | 0.138 |
| | 40 | 0.129 | 1.639 | 0.392 | 0.266 | 0.162 | 0.037 | 0.301 | 0.020 | 1.265 | 0.277 | 0.144 |
| | 50 | 0.187 | 0.578 | 0.391 | 0.183 | 0.090 | 0.037 | 0.226 | 0.026 | 1.428 | 0.196 | 0.076 |
| Maqu (semi-humid) | 5 | 0.277 | 39.041 | 0.790 | 0.750 | 0.287 | 0.047 | 0.766 | 0.016 | 1.329 | 0.611 | 0.301 |
| | 10 | 0.253 | 39.165 | 0.724 | 0.695 | 0.288 | 0.047 | 0.600 | 0.011 | 1.268 | 0.576 | 0.296 |
| | 20 | 0.241 | 37.894 | 0.660 | 0.645 | 0.272 | 0.047 | 0.543 | 0.012 | 1.251 | 0.530 | 0.284 |
| | 40 | 0.199 | 33.131 | 0.535 | 0.534 | 0.250 | 0.047 | 0.472 | 0.013 | 1.225 | 0.454 | 0.254 |
| | 80 | 0.268 | 36.610 | 0.558 | 0.556 | 0.206 | 0.047 | 0.493 | 0.014 | 1.314 | 0.488 | 0.208 |





**Figures**

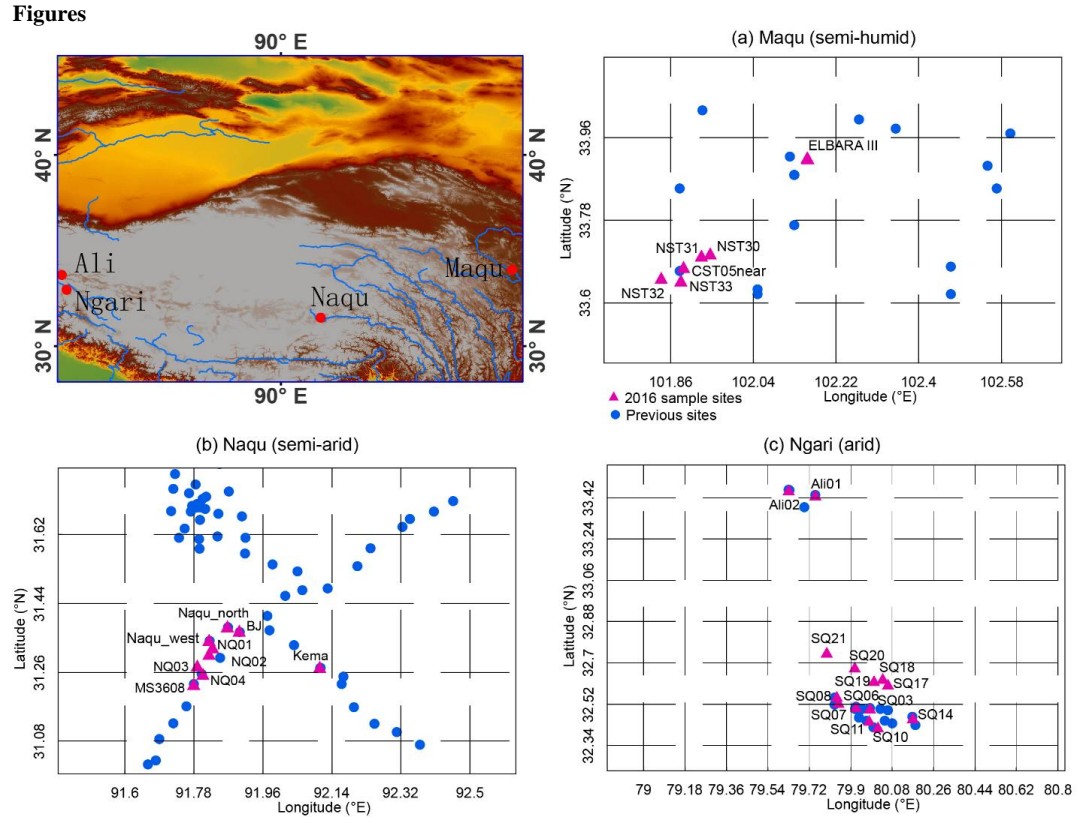


Figure 1 Location of Tibet-Obs and the spatial distribution of soil sampling across three climate zones.

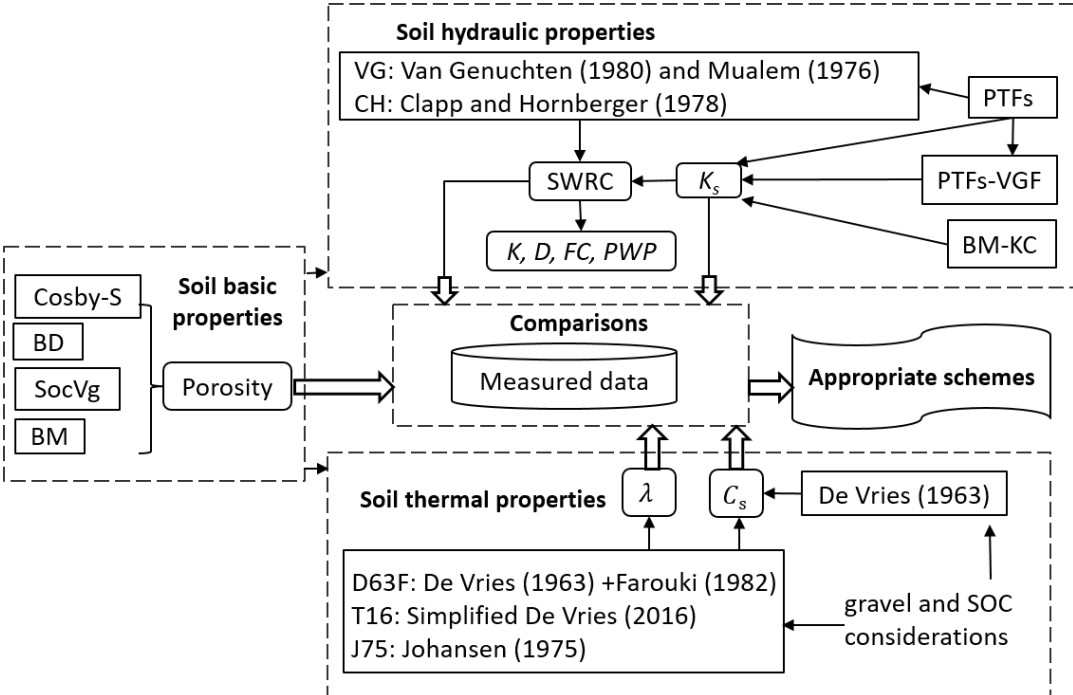

**Figure 2 Flowchart depicting the implementation of different schemes of porosity and SHP/STP. Dashed boxes indicate various categories of parameterization schemes and the comparisons with *in situ* measurements. Block arrows show the main data flow for comparisons. Single arrows represent the steps that occur internally for each part or connect various parts. Rectangles represent schemes. Rounded rectangles denote porosity and SHP/STP parameters. K and D represent hydraulic conductivity and diffusivity, respectively.**




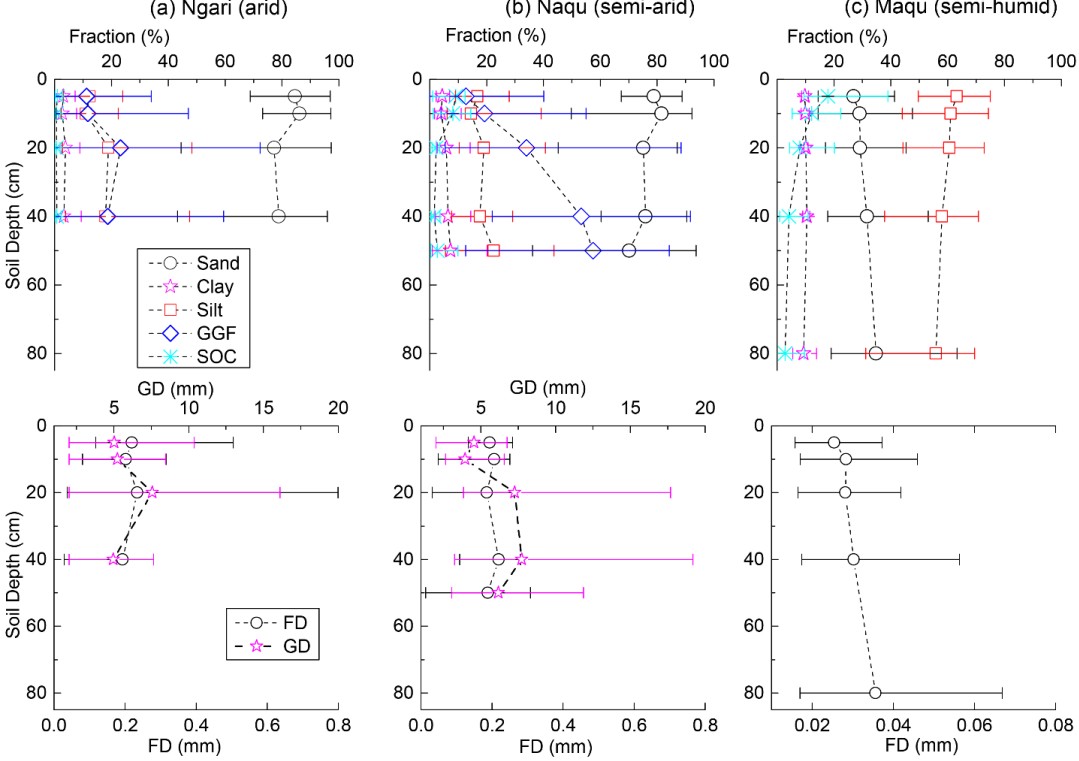

**Figure 3 Profiles of soil basic properties at three climate zones. Top panel: Variations in Sand, Clay, Silt, GGF, and SOC at various depths. Bottom panel: Variations in GD and FD at different depths. GGF is the gravimetric gravel fraction. SOC is the soil organic matter content. FD is the mean particle diameter of fine minerals. GD is the mean particle diameter of gravels. The bar represents the lowest and highest values in the profile.**

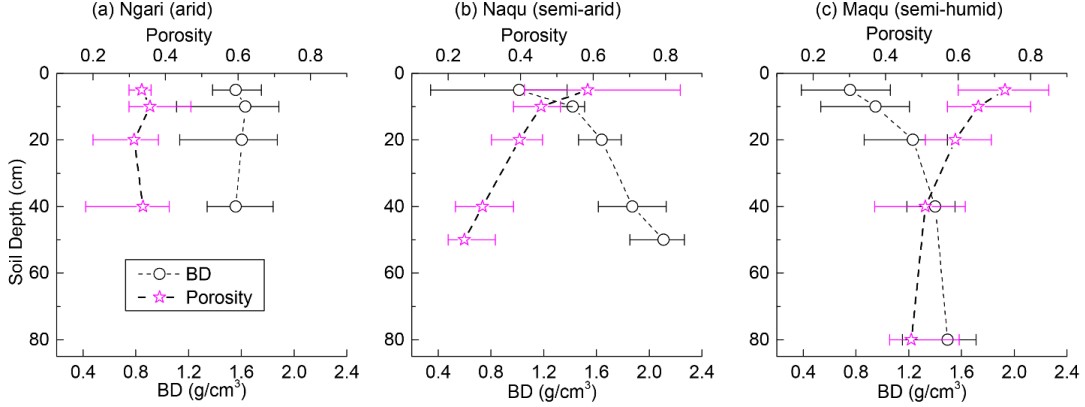


**Figure 4 Profiles of bulk density (BD) and porosity at three climate zones. The bar represents the lowest and highest values in the profile.**



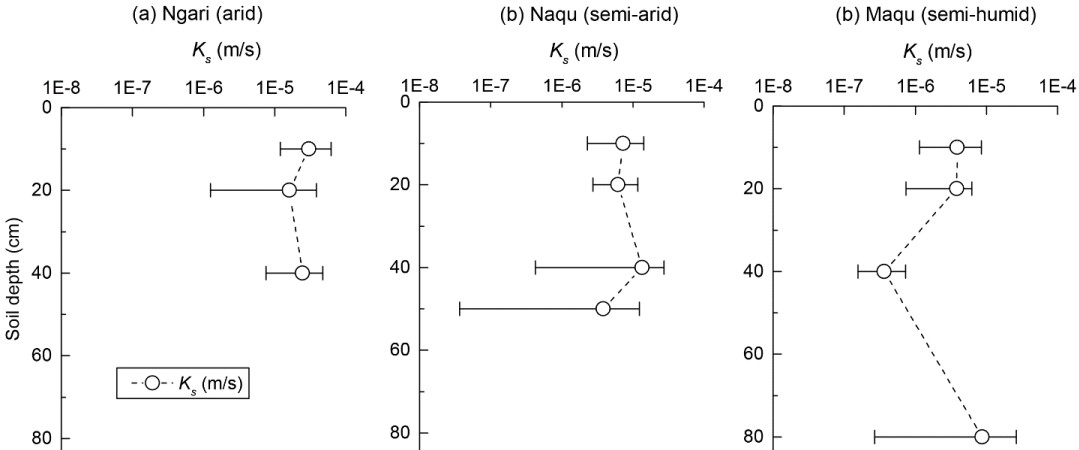

**Figure 5 Profiles of saturated hydraulic conductivity (Ks) at three different climate zones. The bar represents the lowest and highest values in the profile.**

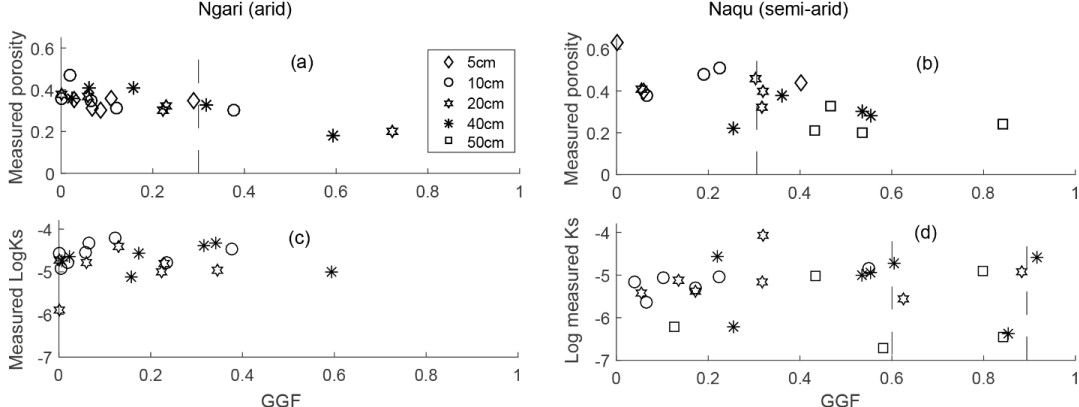

**Figure 6   Scatter points of measured porosities (Top panel) and Ks (Bottom panel) with GGF varying at profile in the arid and semi-arid zones.**



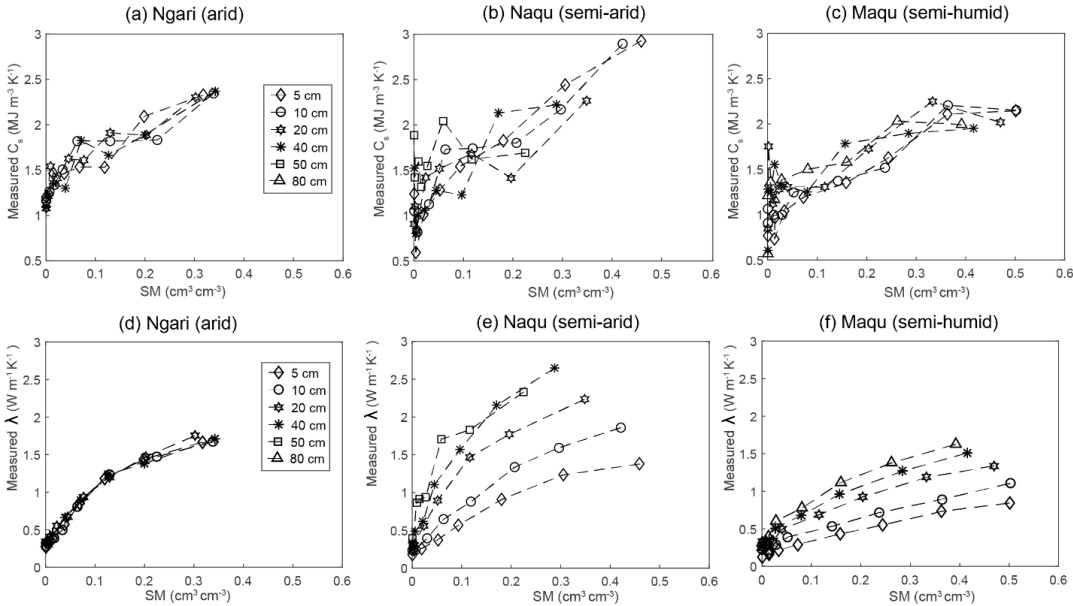

**Figure 7 Profiles of soil heat capacity (Cs) and thermal conductivities ($\lambda$) with varying water content (SM) in three climate zones.**

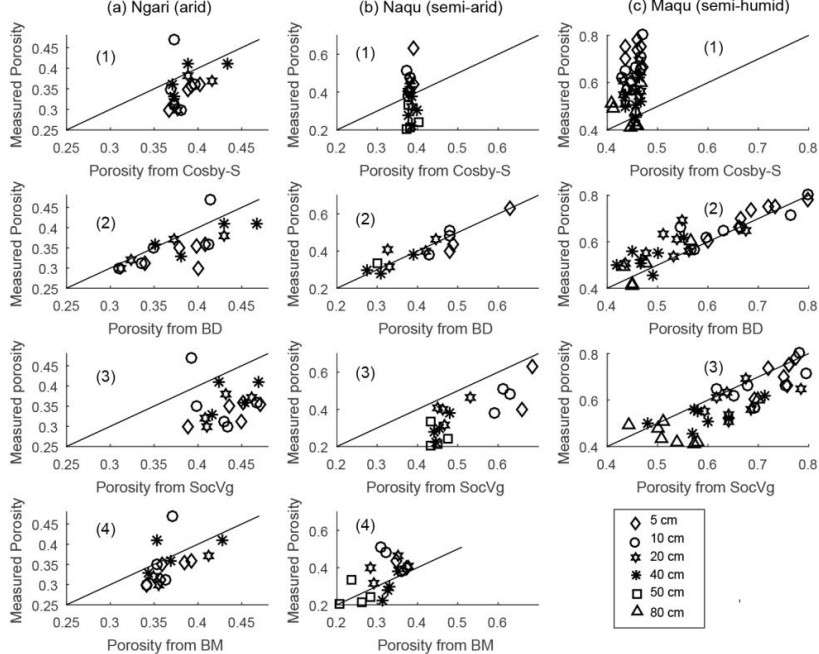

**Figure 8 Comparisons of porosity profiles derived from different schemes with those measured for the three climate zones.**



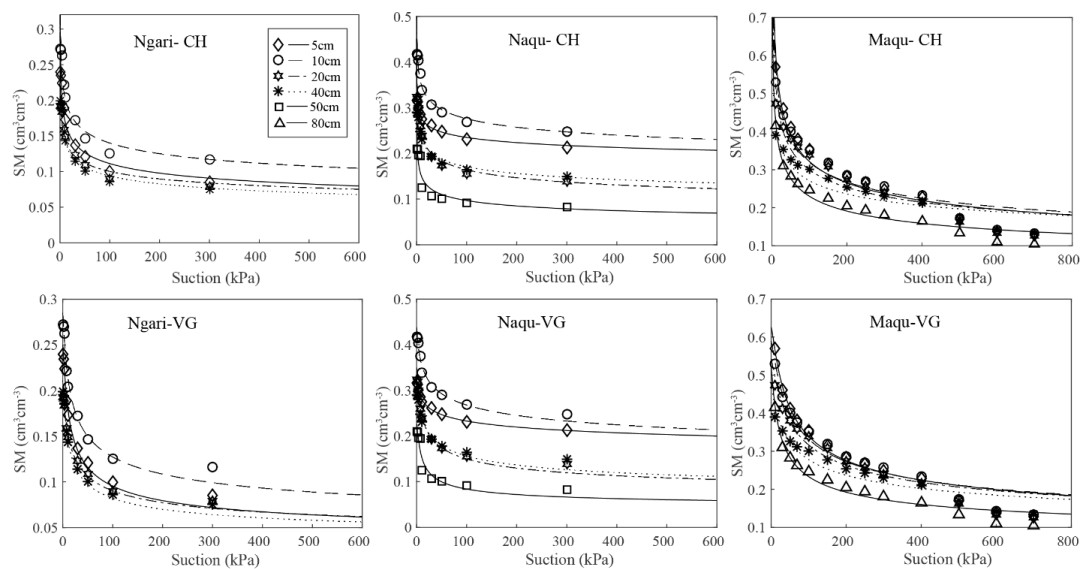


**Figure 9 Estimated SWRCs by the CH and VG models with average observational SWCC of the soil profile for the three climate zones: Ngari in the arid zone, Naqu in the semi-arid zone and Maqu in the semi-humid zone. Dots denote average observed soil moisture content at specific suction. Lines represent estimated SWRCs.**








**Figure 10 Comparisons between mean estimated SWRCs from PTFs combined with the BD scheme, and the laboratory measurements for three climate zones. The broken green line in the right-hand figure of (c) denotes SM at > 100 kPa being above 0.7 cm3cm3.The bar represents the lowest and highest values of the profile.**






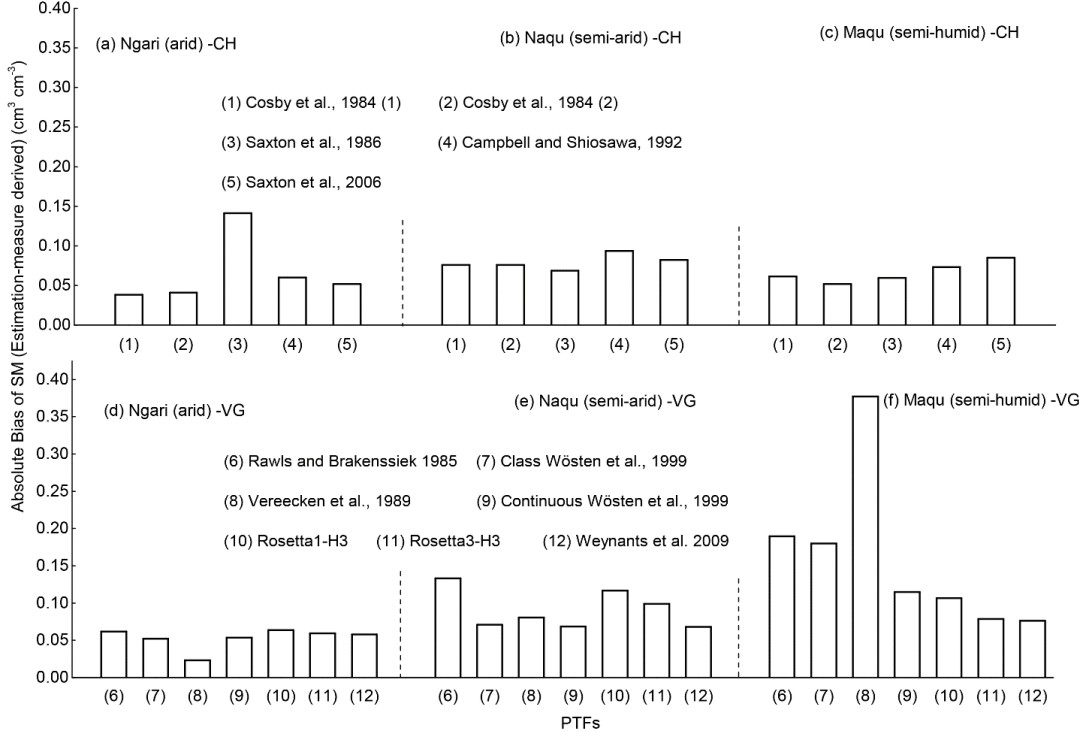

**Figure 11 Absolute bias of estimated SWRCs from PTFs combined with the BD scheme and the measurements at three climate zones.**



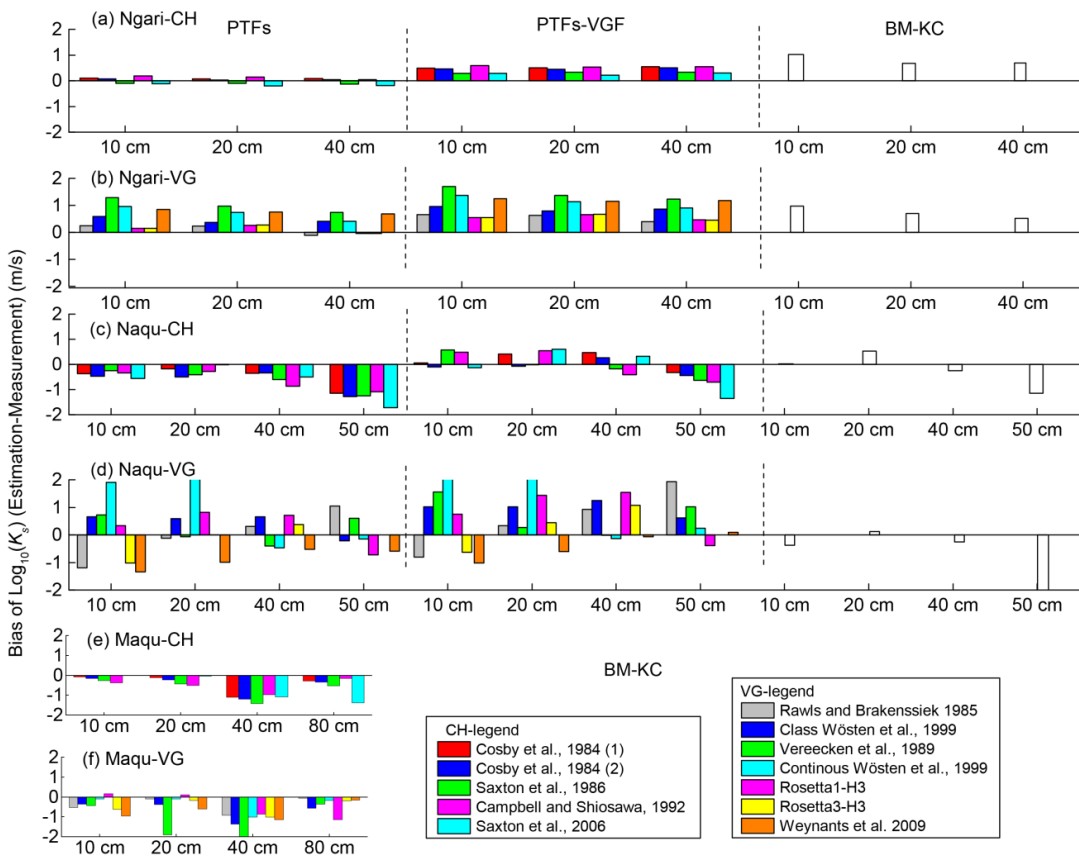

**Figure 12 Comparisons of Ks, derived from PTFs, PTFs-VGF and BM-KC schemes in the CH and VG models, with field measurements in the profile over three climate zones.**

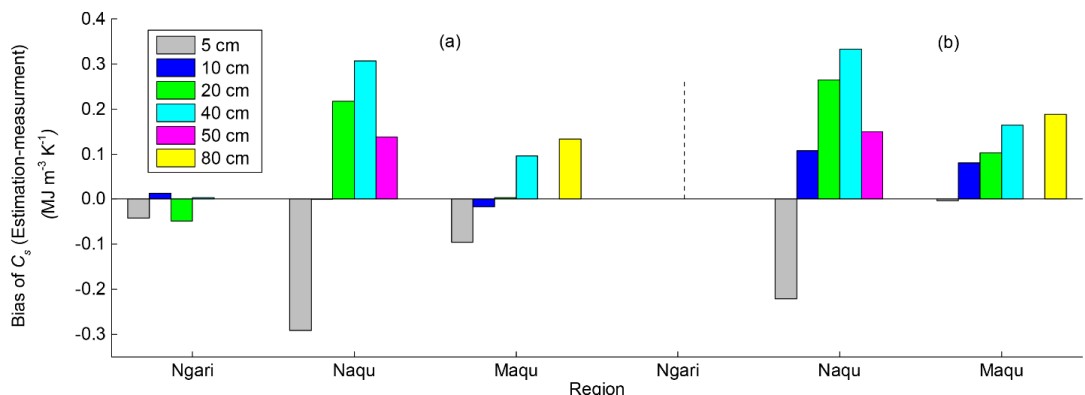

**Figure 13 Biases of Cs between estimates, based on the De Vries (1963) model across the three climate zones, and the measurements. (a) and (b) represent the bias derived from estimations without and with considering SOC impact parameterization in the Maqu and Naqu regions. Ngari denotes the arid zone, Naqu the semi-arid zone and Maqu the semi-humid zone.**



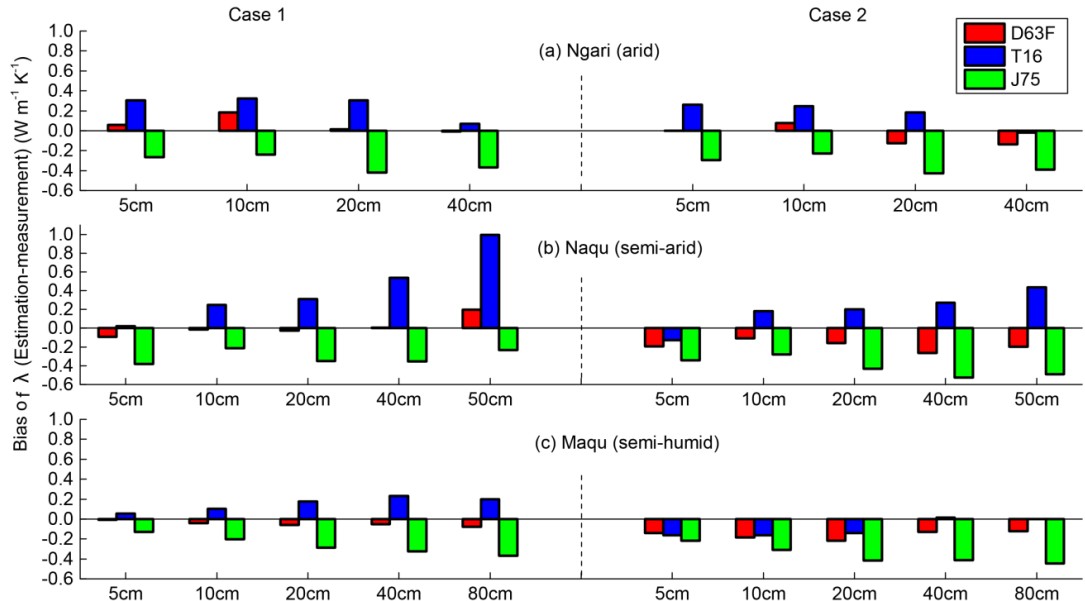

**Figure 14 Biases of λ estimates based on D63F, T16 and J75 schemes combined with the BD scheme in the profile profiles across the three climate zones and the measurements. Case 1 is the bias derived from schemes not considering gravel impact parameterization for the arid and semi-arid zone or SOC impact parameterization for the semi-humid zone. Case 2 is the bias with these parameterizations taken into consideration.**




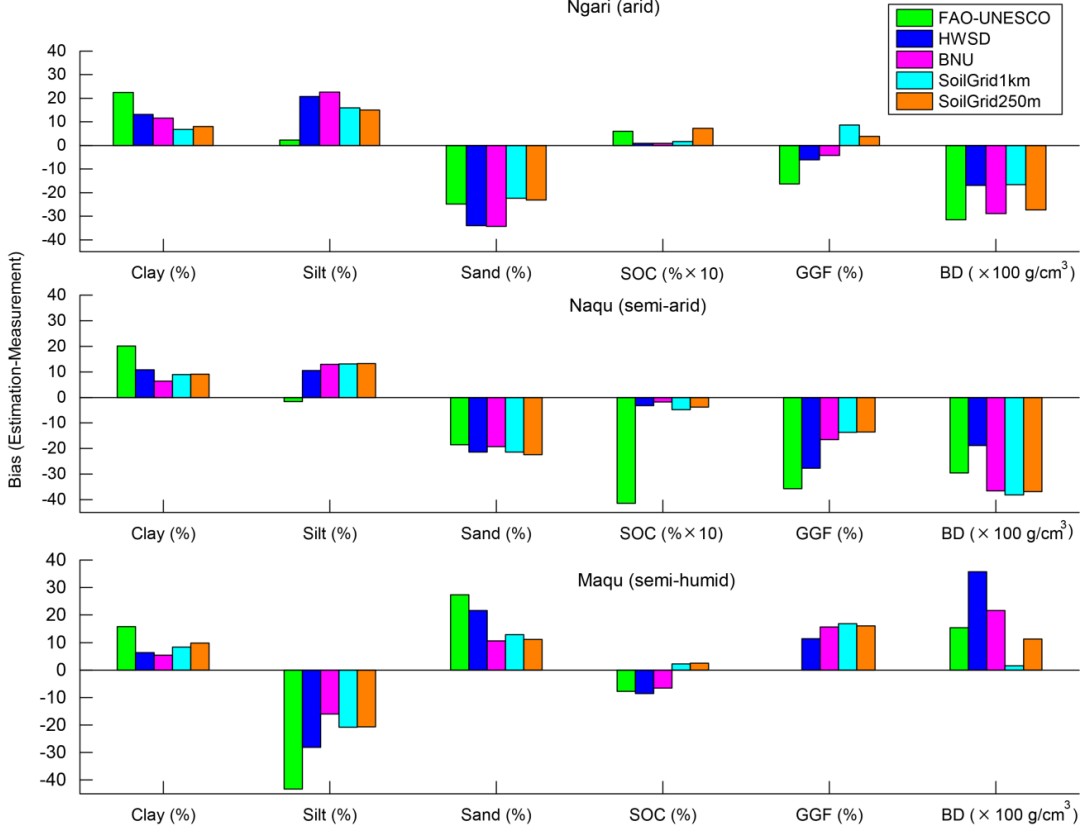

**Figure 15 Average bias in soil basic properties between the existing products and the laboratory measurements taken for the three climate zones. To enable the comparison of BD with the same order of magnitude as other properties, the original BD was multiplied by 100 (with unit × 100 g/cm3). Likewise, a multiplication (% × 10) is applied to SOC data on the semi-arid zone.**

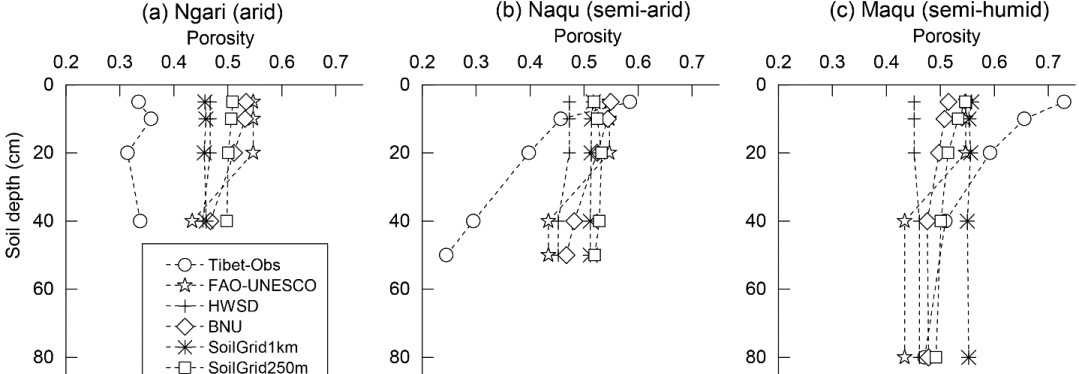

**Figure 16 Comparisons between the porosity estimated from various existing datasets based on BD scheme, and the *in situ* measurements.**





**Figure 17 Comparisons of SWRCs, derived from the applicable PTFs based on various datasets, with laboratory measurements. The bar represents the lowest and highest SM at a specific suction. The left panels represent the SWRCs by the CH model. The right-hand panels represent the SWRCs by the VG model.**




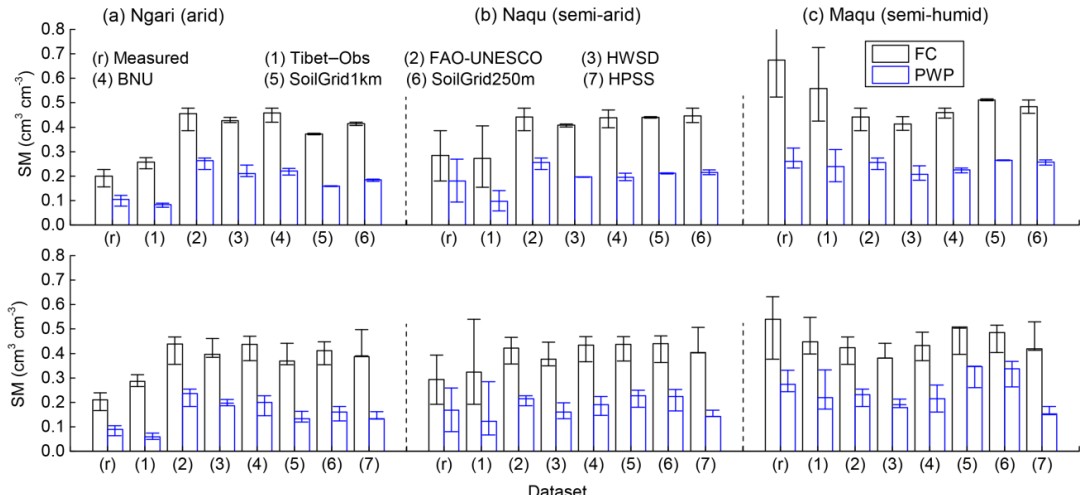

**Figure 18 Comparisons of derived FC and PWP from SWRCs by the CH (upper panel) and VG (lower panel) models based on various soil datasets, with the laboratory measurements. The bar represents the lowest and highest FC and PWP of the profile. FC represents field capacity, and PWP denotes permanent wilting point.**

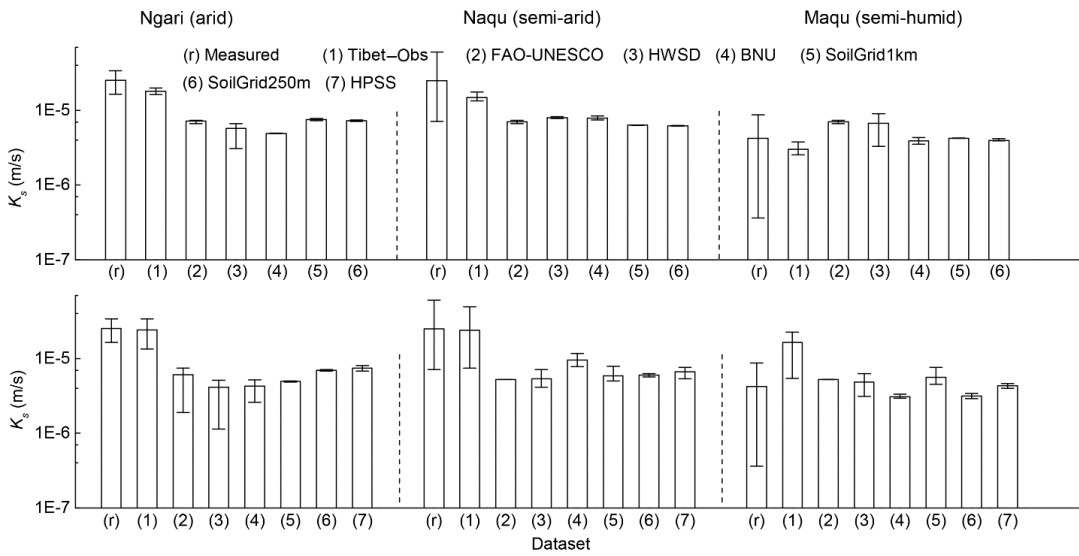


**Figure 19 Comparisons of derived Ks from the applicable PTFs for the CH (upper panel) and VG (lower panel) models based on various soil datasets, with the measurements. The bars represent the lowest and highest Ks in the profile.**





Figure 20 Comparisons of derived soil conductivity (K) and soil diffusivity (D) by the CH model based on various soil datasets, with those derived from the laboratory measurements. The bar represents the lowest and highest K and D at specific soil water content in the profile.




**Figure 21 Comparisons of derived soil conductivity (K) and soil diffusivity (D) by the VG model based on various soil datasets, with those derived from the laboratory measurements. The bar represents the lowest and highest K and D at specific soil water content in the profile.**





**Figure 22 Comparisons of averaged Cs -SM (left panel) and $\lambda$-SM (right-panel) derived from the D63F model based on various datasets, with the measurements. The bar represents the lowest and highest Cs and $\lambda$ in the profile.**



## Appendix

### A.1 Porosity scheme

*Cosby-S scheme*

Cosby et al. (1984) PTF is used to obtain porosity from sand percentage in soil texture:

$$\phi = 0.489 - 0.001268 \times (\%\text{sand}) \qquad (1)$$

where $\phi$ is the soil porosity, *%sand* is the sand proportion of the soil sample.

### BD scheme

BD scheme for porosity calculation (Hillel, 2003) is as follows:

$$\phi = 1 - \frac{\rho_b}{\rho_s} \qquad (2)$$

where $\rho_b$ is the bulk density (g/cm³). $\rho_s$ is the mineral particle density valued at 2.65g/cm³. For soil mixture, BD scheme assumed that the coarse and fine components share the same particle density.

*SocVg scheme*

Regarding soils as a mixture of organic and fine minerals, Chen et al. (2012) conceptualized porosity as shown in Eq. (3). Through the determination of volumetric SOC, the gravel impact was taken into account (Eq. (4-5)) and assumed to be equal to the impact from sand particles. The effective sand proportion was equal to Eq. (6).

$$\phi_m = (1 - V_{soc})\phi_F + V_{soc}\phi_{soc,sat} \qquad (3)$$

$$V_{soc} = \frac{\rho_s(1-\phi_F)m_{soc}}{\rho_{soc}(1-m_{soc})+\rho_s(1-\phi_F)m_{soc}+(1-\phi_F)\frac{\rho_{soc}GGF}{(1-GGF)}} \qquad (4)$$

$$\text{VGF} = \frac{\rho_{soc}(1-\phi_F)\text{GGF}}{(1-GGF)(\rho_{soc}(1-m_{soc})+\rho_s(1-\phi_F)m_{soc}+(1-\phi_F)\frac{\rho_{soc}GGF}{(1-GGF)})} \qquad (5)$$

$$\%sand_e = \%sand * (1 - VGF) + VGF \qquad (6)$$

where $\phi_m$ is the porosity of soil mixture. $V_{soc}$ and $VGF$ are the volumetric fractions of SOC and gravel, respectively.
$\phi_F$ is the porosity of fine components and was calculated by using Eq. (1), where %sand was obtained from Eq. (6). $GGF$ and $m_{soc}$ are the gravimetric fractions of gravels and SOC, respectively. $\rho_{soc}$=0.13 g/cm³ is the BD of peat. $\phi_{soc,sat}$=0.9 is the porosity of peat.

### Binary mixture (BM) scheme

Zhang et al. (2011) proposed a mixing-coefficient model to estimate the porosity for binary mixtures:

$$\phi_m = \begin{cases} (VGF - \beta_m * VGF + \beta_m)\phi_g + VFF * \phi_F - \beta_m * VFF & if\ VFF < \phi_g \\ (1 - \beta_m) * VGF * \phi_g + VFF * \phi_F & if\ VFF \geq \phi_g \end{cases} \qquad (7)$$

where $VFF$ is the component fraction by volume for fine minerals. $VGF$ can be determined using Eq. (8). $\phi_F$ is defined as in the SocVg scheme. $\phi_g$ is the porosity for gravels, which is mainly affected by median grain size (Frings et al., 2011). In this study, $\phi_g$ was calculated by using empirical Eq. (9) given by Wu and Wang (2006). $\beta_m$ is the mixing coefficient related with grain size (Eq. (10)).

$$VGF = \frac{GGF(1-\phi_F)}{GGF(1-\phi_F)+(1-GGF)(1-\phi_g)} \qquad (8)$$





$$\phi_g = 0.13 + \frac{0.21}{(GD+0.002)^{0.21}} \qquad (9)$$

$$\beta_m = \begin{cases} 0.0363\frac{GD}{FD} + 0.2326 & for \frac{GD}{FD} \leq 21 \\ 1 & for \frac{GD}{FD} > 21 \end{cases} \qquad (10)$$

where $GD$ and $FD$ are the mean grain size for gravels and fine minerals, respectively, and the unit is millimeter.

**A.2 Soil water retention curve (SWRC)**

The function of Clapp and Hornberger (1978) (i.e. CH) for soil water retention is written as:

$$\varphi = \varphi_s(\theta/\theta_s)^{-1/b} \quad \varphi \leq \varphi_i \qquad (11)$$

where $\varphi_s$ is the saturated capillary potential (cm). $b$ is pore size distribution index (dimensionless). $\theta$ is the SM (cm$^3$cm$^{-3}$) and $\theta_s$ is the saturated SM. $\varphi_i$ defines an inflection point near saturation. The soil conductivity and diffusivity are written as:

$$\begin{cases} K(\theta) = K_s \left(\theta/\theta_s\right)^{3+2/b} \\ D(\theta) = D_s \left(\theta/\theta_s\right)^{2+1/b} \\ D_s = 1/b * K_s(\varphi_s/\theta_s) \end{cases} \qquad (12)$$

where $K$ and $D$ are the soil hydraulic and thermal conductivity. $K_s$ and $D_s$ are the saturated hydraulic conductivity (m/s) and diffusivity (m$^2$/s).

The van Genuchten (1980) and Mualem (1976) (i.e. VG) model provides the water retention curve as Eq. (13) shows,

$$\theta(\boldsymbol{h}) = \theta_r + \frac{\theta_s-\theta_r}{(1+(ah)^n)^{1-1/n}} = f(\boldsymbol{h}, \theta_r, \theta_s, \alpha, n) \qquad (13)$$

where $\theta(h)$ is the SM (cm$^3$cm$^{-3}$) at pressure head $h$ (cm). $\theta_r$ is the residual SM (cm$^3$cm$^{-3}$). $\theta_s$ has the same meaning as above. $\alpha$ is the inverse of air entry value (cm$^{-1}$). $n$ is the shape parameter (dimensionless). The soil conductivity and diffusivity are written as:

$$\Theta = \frac{\theta - \theta_r}{\theta_s - \theta_r}$$

$$K = K_s \Theta^{1/2}\left[1 - (1 - \Theta^{1/(1-1/n)})^{1-1/n}\right]^2 \qquad (14)$$

$$D(\Theta) = \frac{(1-m)K_s}{\alpha m(\theta_s - \theta_r)}\Theta^{1/2-1/m}\left[\left(1-\Theta^{1/m}\right)^{-m} + \left(1-\Theta^{1/m}\right)^m - 2\right]$$

$$m = 1 - 1/n$$

where $\Theta$ is the effective saturation.

Based on measured soil water potential and SM, we adopted the scaling method proposed by Montzka et al. (2017)
to estimate hydraulic parameters in the CH and VG models. The expected-scale (representative) parameters of $(\widehat{\theta_s}, \hat{b}, \widehat{\varphi_s})$ and $(\widehat{\theta_r}, \widehat{\theta_s}, \hat{\alpha}, \hat{n})$ in the water retention curves of $f(h, \theta_s, b, \varphi_s)$ and $f(h, \theta_r, \theta_s, \alpha, n)$ that minimize the sum of squares of the deviations for all respective observations $i = 1 \dots N$ (Eq. (14)) need to be obtained. The parameter fitting algorithm was the damped least-squares method of Levenberg-Marquardt (Marquardt, 1963). The initial values were taken from the mean of $(\theta_s, \alpha, n)$ and $(\theta_r, \theta_s, \alpha, n)$ at each observation.

$$\left(\widehat{\theta_s}, \hat{b}, \widehat{\varphi_s}\right) = \text{argmin} \sum_{i=1}^{N} [\theta_i - f(\boldsymbol{h}, \theta_{s,i}, b_i, \varphi_{s_i})]^2$$

$$\left(\widehat{\theta_r}, \widehat{\theta_s}, \hat{\alpha}, \hat{n}\right) = \text{argmin} \sum_{i=1}^{N} [\theta_i - f(\boldsymbol{h}, \theta_{r,i}, \theta_{s,i}, \alpha_i, n_i)]^2 \qquad (14)$$





### A.3 PTFs for SWRC

Various PTFs have been developed to determine soil hydraulic properties. In terms of criterions described in Dai et al. (2013), five PTFs (No. 1-5 in Table A1) were selected for estimating parameters of ($\theta_s$, $\varphi_s$, $b$) in the CH model as well as seven PTFs (No. 6-12 in Table A1) for the parameters of ($\theta_r$, $\theta_s$, α, n) in the VG model.


*Table A1 List of PTFs for estimating soil water retention curve.*

| # | PTF | Retention/ $K_s$ model | Sand % | Silt % | Clay % | Organic Carbon % | Bulk density g cm$^{-3}$ | Depth - |
|---|-----|------|------|------|------|------|------|------|
| 1 | Cosby et al., 1984 (1) | CH, $K_s{}^1$ | √ | | √ | | | |
| 2 | Cosby et al., 1984 (2) | CH, $K_s{}^2$ | √ | √ | √ | | | |
| 3 | Saxton et al., 1986 | CH, $K_s{}^3$ | √ | | √ | √ | | |
| 4 | Campbell and Shiosawa, 1992 | CH, $K_s{}^4$ | √ | √ | √ | | √ | |
| 5 | Saxton et al., 2006 | CH, $K_s{}^5$ | √ | | √ | √ | | |
| 6 | Rawls and Brakenssiek 1985 | VG, $K_s{}^6$ | √ | | √ | | √ | |
| 7 | Class Wösten et al., 1999 | VG, $K_s{}^7$ | √ | √ | √ | | | √ |
| 8 | Vereecken et al., 1989 | VG, $K_s{}^8$ | √ | | √ | √ | √ | |
| 9 | Continuous Wösten et al., 1999 | VG, $K_s{}^9$ | | √ | √ | √ | √ | √ |
| 10 | Rosetta1-H3 | VG, $K_s{}^{10}$ | √ | √ | √ | | √ | |
| 11 | Rosetta3-H3 | VG, $K_s{}^{11}$ | √ | √ | √ | | √ | |
| 12 | Weynants et al. 2009 | VG, $K_s{}^{12}$ | √ | | √ | √ | √ | |

where 10 and 11 were cited by Schaap et al. (2001) and Zhang and Schaap (2017), respectively.

### A.4 Saturated hydraulic conductivity scheme
*PTFs-VGF scheme*


The PTF-VGF scheme estimated $K_s$ of soil mixtures (Peck and Watson, 1979) as follows:

---

[1] $K_s = 60.96 * 10^{-0.884+0.0153*sand}$

[2] $K_s = 60.96 * 10^{-0.6+0.0126*sand-0.0064*clay}$

[3] $K_s = 24.0\exp\{12.012 - 0.0755*sand + [-3.895 + 0.03671*sand - 0.1103*clay + 0.00087546clay^2]/\theta_s\}$

[4] $K_s = 339.0 * \left(\frac{1.3}{BD}\right)^{1.3b} \exp(-0.06888*clay - 0.03638*silt - 0.025)$

[5] $x = 0.00251 * sand + 0.00195 * clay + 0.011 * SOC + 0.00006 * sand * SOC\ 0.00027 * clay * SOC + 0.0000452 * sand.* clay + 0.299$; $K_s = 4632 * (\theta_s - y)^{3-b}$

[6] $\theta_s = \phi = 1 - BD/2.65$; $K_s = 24\exp(19.52348 * \phi - 8.96847 - 0.028212 * clay + 0.00018107 * sand^2 - 0.0094125 * clay^2 - 8.395215 * \phi^2 + 0.077718 * sand * \phi - 0.00298 * sand^2 * \phi^2 - 0.019492 * clay^2 * \phi^2 + 0.0000173 * sand^2 * clay + 0.02733 * clay^2 * \phi + 0.001434 * sand^2 * \phi - 0.0000035 * sand * clay^2$;

[7] The $K_s$ for the FAO textural classes Pachepsky, Y., and Rawls, W. J.: Development of pedotransfer functions in soil hydrology, Elsevier, 2004.

[8] $\text{Log}(K_s) = 20.62 - 0.96 * \log(clay) - 0.66 * \log(sand) - 0.46 * \log(clay) - 8.43 * BD$

[9] $K_s = \exp(7.75 + 0.0352 * silt + 0.93 * itop - 0.967 * BD^2 - 0.000484 * clay^2 - 0.000322 * silt^2 + 0.001/silt - 0.0748/SOC - 0.643 * \log(silt) - 0.01398 * BD * clay - 0.1673 * BD * SOC + 0.02986 * itop * clay - 0.03305 * itop * silt))$, where topsoil is an ordinal variable having the value of 1 (depth 0–30 cm) or 0 (depth 30 cm).

[10] H3 hierarchical pedotransfer function in Schaap et al. (2001)

[11] Updated H3 hierarchical pedotransfer function in Zhang, Y., and Schaap, M. G.: Weighted recalibration of the Rosetta pedotransfer model with improved estimates of hydraulic parameter distributions and summary statistics (Rosetta3), J. Hydrol., 547, 39-53,doi: 10.1016/j.jhydrol.2017.01.004, 2017.

[12] $K_s = \exp(1.9582 + 0.0308sand - 0.6142BD - 0.01566SOC * 1.72)$





$$K_{sm} = K_{sat,f} \frac{2(1-VGF)}{2+VGF} \qquad (15)$$

where $K_{sm}$ is the $K_s$ of soil mixtures. $K_{sat,f}$ is the $K_s$ of fine minerals and was calculated using PTFs in Table A1. VGF shares the same definition as for Eq. (8).

### BM-Kozeny-Carman equation (BM-KC scheme)

The Kozeny-Carman equation (16), originally developed to quantitatively describe hydraulic conductivity vs. the mean grain size in capillary flow, was used to estimate $K_s$ of binary mixtures. The porosity was obtained by using the BM scheme in A.1 section. The representative grain diameter was estimated using the power-averaging method (Eq. (16)) proposed by Zhang et al. (2011). This method introduced a coefficient (Eq. (17)) with the critical fraction of gravels taking into account.


$$K_{sm} = \left(\frac{\rho g}{\mu}\right)\left[\frac{d_m^2 \phi_m^3}{180(1-\phi_m)^2}\right] \qquad (16)$$

where $\phi_m$ has the same definition as in Equation (7). $d_m$ is the representative grain diameter of soil mixture. ρ is the fluid density. $g$ is gravitational acceleration, and μ is the dynamic viscosity.

$$d_m = \left(VGF * GD^p + VFF * FD^p\right)^{1/p} \quad (17)$$

where VGF, VFF, GD and FD have the same definition as in the BM scheme in A.1 section displays. $p$ is a
coefficient that varies sigmoidally from −1 to 0 with VGF increasing from 0 to 1. $p$ is estimated empirically by

$$p = \frac{1}{1+\exp[(\alpha(VGFc-VGF))]} - 1 \qquad (18)$$

where $VGFc$ is the critical fraction of gravels and is approximated by $VGFc = 1 - \phi_g$ ($\phi_g$ from Eq. (9)). $\alpha$ is a shape factor set at a value of 20 as in Zhang et al. (2011).

### A.5 Heat capacity and thermal conductivity
*Heat capacity*

Soil heat capacity $C_s$ depends on the heat capacities of all constituents, and is calculated using Eq. (19) given by De Vries (1963),

$$C_s = \theta C_w + (1-\theta_s)C_{soil} + (\theta_s - \theta)C_{air} \qquad (19)$$

where $\theta$ and $\theta_s$ share the same meaning as in Eq. (11). $C$ represents the heat capacity (MJ m$^{-3}$ K$^{-1}$), and the
subscripts 'w', 'soil' and 'air' refer to water, solid soil and air, respectively. $C_w$, $C_{soil}$ and $C_{air}$ are taken as 4.2, 2.0 and 0.001 MJ m$^{-3}$ K$^{-1}$, respectively. If taking SOC impact into consideration, $C_s$ is calculated as Eq. (20) shows,

$$C_s = \theta C_w + (1-\theta_s) * ((1-Vsoc)*C_{soil} + V_{soc}*C_{soc}) + (\theta_s - \theta)C_{air} \qquad (20)$$

where $Vsoc$ shares the same definition as in Eq. (4). $C_{soc}$ is the heat capacity of organic matter and taken as 2.5 MJ m$^{-3}$ K$^{-1}$.

*Thermal conductivity by the De Vries (1963) model revised by Farouki (1981) (D63F)*

The De Vries (1963) model was developed from the Maxwell equation for electrical conductivity of a mixture of granular materials dispersed in a continuous fluid (Eucken, 1932). Farouki (1981) set liquid water as the continuous medium and regarded soil minerals as uniform particles. Considering soils as binary mixture of fine minerals and coarse gravels, $\lambda$ is estimated as follows:


$$\lambda = \frac{x_w \lambda_w + w_a x_a (\lambda_a + \lambda_v) + w_m x_m \lambda_m + w_g x_g \lambda_g + w_{soc} x_{soc} \lambda_{soc}}{x_w + w_a x_a + w_m x_m + w_g x_g + w_{soc} x_{soc}} \quad (21)$$





where $w$ is the weighting factor, $x$ is the volume fraction, $\lambda$ is the thermal conductivity, and the subscripts 'w', 'a', 'v', 'm', 'g' and 'soc' refer to water, air, vapor, fine minerals, gravels and SOC composed of soil, respectively. $\lambda_w$ =0.57 W m⁻¹ K⁻¹, $\lambda_a$ =2.0 W m⁻¹ K⁻¹, $\lambda_g$ =2.54 W m⁻¹ K⁻¹ and $\lambda_{soc}$ =0.25 W m⁻¹ K⁻¹. $\lambda_m$ was calculated using Eq. (22),

$$\lambda_m = \lambda_q^q \lambda_o^{(1-q)} \quad (22)$$

where $\lambda_q$ is the thermal conductivity of quartz ($\lambda_q$=7.7 W m⁻¹ K⁻¹). $\lambda_o$ is the thermal conductivity of other minerals ($\lambda_o$=2.0 W m⁻¹ K⁻¹). $Vsoc$ shares the same meaning as for Eq. (4). In this study, $q$ is assumed equal to half of the sand fraction ($q = 1/2\%sand$) in terms of Chen et al. (2012)'s research.

$w$ in Eq. (23) is empirically given by:

$$w_i = \frac{1}{3}\left[\frac{2}{1+\left(\frac{\lambda_i}{\lambda_w}-1\right)g_a} + \frac{1}{1+\left(\frac{\lambda_i}{\lambda_w}-1\right)(1-2g_a)}\right] \quad (23)$$

where $g_a$ is the shape factor of ellipsoidal particles. A uniform shape factor $g_a$ of 0.125 is used for fine minerals (Farouki, 1981), a $g_a$ of 0.33 for gravels and a $g_a$ of 0.5 for SOC (De Vries, 1963).

For $\lambda_v$ together with $g_a$ for air, Farouki (1981) provided the following Eq. (24).

For 0.09 m³ m⁻³ $\leq x_w \leq \phi$,

$\lambda_v = \lambda_v^s$   and   $g_{a(air)} = 0.333 - (0.333 - 0.035)x_a/\phi$

And for $0 \leq x_w \leq 0.09$ m³ m⁻³, (24)

$\lambda_v = \frac{x_w}{0.09}\lambda_v^s$   and   $g_{a(air)} = 0.013 + 0.944\phi$

where $\lambda_v^s$ is the value of $\lambda_v$ for saturated vapor. $\phi$ is defined as in Eq. (2).

***Simplified De Vries-based model (T16)***

The T16 scheme (Tian et al., 2016) assumed the negligible effect of vapor movement (i.e. $\lambda_v = 0$) in the De Vries-based model (Eq. (19)). Soil texture was assumed to be determining the physical properties of soil minerals. $\lambda$ of fine minerals ($\lambda_m$) and shape parameters for minerals and air were computed using Eq. (25-27),

$$\lambda_m = \lambda_{sand}^{\%sand}\lambda_{clay}^{\%clay}\lambda_{silt}^{\%silt} \quad (25)$$

and   $g_{a(m)} = g_{a(sand)}\%sand + g_{a(silt)}\%silt + g_{a(clay)}\%clay$ (26)

where $\lambda_{sand}$=7.7 W m⁻¹ K⁻¹, $g_{a(sand)}$=0.782, $\lambda_{silt}$=2.74 W m⁻¹ K⁻¹, $g_{a(silt)}$=0.0534, $\lambda_{clay}$=1.93 W m⁻¹ K⁻¹, and $g_{a(clay)}$=0.00775.

$g_{a(air)}$ is assumed to vary linearly with air fraction and is estimated using Eq. (27):

$$g_{a(air)} = 0.333 * (1 - x_a/\phi) \quad (27)$$

where $\phi$ is defined as in Eq. (2). $x_a$ has the same meaning as in Eq. (21).

For dry soils, $\lambda_{dry}$ calculation follows Eq. (26) proposed by De Vries (1963),

$$\lambda_{dry} = 1.25 * \frac{w_a x_a \lambda_a + w_m x_m \lambda_m + w_g x_g \lambda_g + w_{soc}x_{soc}\lambda_{soc}}{w_a x_a + w_m x_m + w_g x_g + w_{soc}x_{soc}} \quad (28)$$

where parameters share the same definition as in Eq. (21).



***Johansen model (J75)***

The Johansen (1975) model simulated $\lambda$ given by a combination of dry and saturated state values, which is weighted
by a factor known as the Kersten number as Eq. (29) depicts,

$$\lambda = K_e\left(\lambda_{sat} - \lambda_{dry}\right) + \lambda_{dry} \qquad (29)$$

where $\lambda_{dry}$ and $\lambda_{sat}$ are the dry and saturated thermal conductivity, respectively. $K_e$ is the Kersten number, a
normalized thermal conductivity that relates to the logarithm of the moisture content (Kersten, 1949) as Eq. (30)
shows,


$$\begin{cases} K_e = \dfrac{\lambda - \lambda_{dry}}{\lambda_{sat} - \lambda_{dry}} \\ K_e = \log(S_r) + 1.0 \quad \text{for unfrozen fine} - \text{grained soils} \\ K_e = 0.7 * \log(S_r) + 1.0 \quad \text{for unfrozen medium and fine sands} \end{cases} \qquad (30)$$

where $S_r$ is the saturation degree and defined as Eq. (31),

$$S_r = {x_w}/{\theta_s} \quad (31)$$

where $x_w$ is the SM (cm$^3$ cm$^{-3}$). $\theta_s$ is saturated SM (cm$^3$ cm$^{-3}$) and calculated using Eq. (2).

The saturated thermal conductivity is calculated using Eq. (32),


$$\lambda_{sat} = \lambda_m^{1-\theta_s}\lambda_w^{\theta_s} \quad (32)$$

where $\lambda_m$ has the same definition as in Eq. (20). If considering the SOC impact, $\lambda_m$ was calculated using Eq. (33),

$$\lambda_m = \lambda_q^{q(1-Vsoc)}\lambda_o^{(1-q)(1-Vsoc)}\lambda_{soc}^{Vsoc} \quad (33)$$

The thermal conductivity for dry state is given as Eq. (34):

$$\lambda_{dry} = \frac{0.135\rho_b + 64.7}{2700 - 0.947\rho_b} \quad (34)$$

where $\rho_b$ is the bulk density (g/cm$^3$).

**Table A2**
Table A2 Bias and RMSE between simulated porosities with measurements at three climate zones

| Scheme | Index | Ngari (arid) | | | | Naqu (semi-arid) | | | | | Maqu (semi-humid) | | | | |
|---|---|---|---|---|---|---|---|---|---|---|---|---|---|---|---|
| | | 5 cm | 10 cm | 20 cm | 40 cm | 5 cm | 10 cm | 20 cm | 40 cm | 50 cm | 5 cm | 10 cm | 20 cm | 40 cm | 80 cm |
| Cosby-S | Bias | 0.05 | 0.02 | 0.07 | 0.05 | 0.18 | 0.08 | 0.05 | 0.09 | 0.14 | 0.27 | 0.20 | 0.14 | 0.07 | 0.06 |
| | RMSE | 0.05 | 0.07 | 0.09 | 0.09 | 0.25 | 0.10 | 0.05 | 0.11 | 0.15 | 0.28 | 0.21 | 0.15 | 0.09 | 0.07 |
| BD | Bias | 0.06 | 0.04 | 0.05 | 0.07 | 0.04 | 0.03 | 0.03 | 0.02 | 0.04 | 0.02 | 0.03 | 0.07 | 0.06 | 0.05 |
| | RMSE | 0.06 | 0.04 | 0.05 | 0.07 | 0.05 | 0.03 | 0.04 | 0.02 | 0.04 | 0.03 | 0.05 | 0.08 | 0.06 | 0.06 |
| SocVg | Bias | 0.1 | 0.07 | 0.09 | 0.11 | 0.14 | 0.15 | 0.08 | 0.16 | 0.20 | 0.04 | 0.06 | 0.07 | 0.08 | 0.08 |
| | RMSE | 0.11 | 0.1 | 0.09 | 0.15 | 0.17 | 0.16 | 0.09 | 0.17 | 0.21 | 0.05 | 0.08 | 0.09 | 0.09 | 0.10 |
| BM | Bias | 0.03 | 0.06 | 0.06 | 0.07 | 0.06 | 0.13 | 0.07 | 0.05 | 0.05 | | | | | |
| | RMSE | 0.03 | 0.06 | 0.06 | 0.07 | 0.07 | 0.15 | 0.08 | 0.06 | 0.06 | | | | | |
