# Peer review of "Analysis of Soil Hydraulic and Thermal Properties for Land Surface Modelling over the Tibetan Plateau"

_Earth System Science Data, 2017_

## Referee Comment (RC1) · A. Verhoef (Referee) · 22 Feb 2018

Review of "Analysis of Soil Hydraulic and Thermal Properties for Land Surface Modelling over the Tibetan Plateau"

By Hong Zhao, Yijian Zeng, Shaoning Lv, Zhongbo Su

This is an interesting paper that presents a wealth of soil physical data and related calculations with semi-empirical theoretical equations (with a focus on hydraulic and thermal soil properties) for the Tibetan plateau. In-situ hydraulic and thermal data are scarce, particularly for thermal properties, so this dataset is a welcome addition to what is available in the literature. As the authors point out, the Tibetan Plateau exerts a huge influence on the Earth's climate system and is important from a land surface-atmosphere feedback point of view, so these data should in theory greatly improve future land surface and climate model outputs for this region and beyond.

**Major comments:**

- Although it is overall easy enough to understand what 'went on', the scientific and English language is imprecise in various places. I have given some examples below, but the authors should go throughout the entire manuscript carefully, and check that their description is as exact as possible.

- They also need to make a bigger effort to refer to the exact equation numbers and Appendix sections when describing their Material and Methods. At the moment this is rather hard to follow for a non-expert.

- How were the SWRCs determined exactly? Were the samples used in the pressure cell method undisturbed and still contained gravel? If this is the case, is it 'fair' to compare the SWRCs of such gravel-rich samples with SWRCs derived from the CH and VG equation with parameters derived from PTFs that were not developed for gravelly soils?

- At times, you have the heat capacity of your (average) samples going up and down with soil moisture content, instead of steadily increasing with SM. What has caused this? This must be an issue with your sensor?

**Minor comments**
Line 41-42: You say: "SHP and STP govern the partitioning of SM between infiltration and evaporation flux and water heat transport processes (Zeng et al., 2009a; Zeng et al., 2009b)", but surely STP only has a significant role on SM during periods of freezing?

Line 44: Should be: 'soil formation factors' or ''soil forming factors'?

Line 44: 'Could be' should be 'can be', also throughout manuscript.

Line 53: What exactly is meant by "soil physical consistency of LSMs "?

Line 69: 'are accessible'

Lines 70-73. I am wondering whether this kind of info is more suitable for the Materials and methods?

Line 92: Could "**2. Experimental Designs and Parameterization Schemes** " not be "2. Materials and methods. Followed by : 2.1 Field experiments, 2.2 Laboratory Experiments, 2.3. Model equations.

Line 101 and throughout: please refer to this soil property as dry bulk density.

Line 120-123: I found this sentence hard to follow: "Based on fine minerals and the standard particle size classes of the United States Department of Agriculture (USDA), sand and clay percentages as well as the mean particle diameter of the fine component (FD) were determined using the Malvern Mastersizer 2000 particle size analyzer". Did you not simply want to say that you determined sand and clay percentages and FD for the fine minerals with the Malvern..?

Line 134-136: Say something like : "only 30 out of 96 samples were used for the Naqu network as the structure of the other samples was so unconsolidated that the material did not remain enclosed within the rings". The same occurred for a number of samples at Maqu and Ngari and therefore samples packed in standard rings were used for Naqu_north, SQ17, SQ18 and SQ21 sites".

Line 136: Were the samples in the standard rings undisturbed? This needs to be made clear. To me 'packed' means that you inserted the soil yourself from bulk samples. Maybe say 'contained in the rings' for undisturbed samples and 'repacked' for disturbed?

Line 138-139: "The quality of the measured soil property dataset was evaluated based on quality indicators (e.g. observation date, level of trust, data quality rating and accuracy)". This sounds somewhat vague? How is the date going to affect the quality, for example?

Line 141: "The entered data (level 'A') have been standardized (level 'B') and harmonized (level 'C')."

Although I appreciate the efforts to ensure data quality and consistency, I would like to know what procedures and criteria are behind the 'standardising' and 'harmonising'. Have you actually changed raw data values??

Line 144-146: You say: "The collected basic soil property and SHP & STP datasets over Tibet-Obs **were further used to evaluate** the existing soil datasets of FAO-UNESCO (FAO/UNESCO, 2007), HWSD (FAO/IIASA/ISRIC/I 145 SSCAS/JR, 2012), BNU (Shangguan et al., 2012; Shangguan et al., 2013), SoilGrid1km (Hengl et al., 2014), SoilGrid250km and HPSS

(Montzka et al., 2017) over the TP. The detailed information of the datasets used is listed in Table S1 of the Supplement".

Although Table S1 lists the details of these databases I can't say that I see an **evaluation** of these datasets using your own data. Did I misunderstand your sentence?

Line 151-152: You say "The Cosby et al. (1984) PTF that used sand percentage (hereafter the Cosby-S scheme) has been widely used".

However, this refers to Cosby's uni-variate equation, but they also present a multi-variate PTF, that uses clay as well as sand. It would be good to mention this. In this Section you could also refer to the in-depth overview of PTFs by Van Looy et al. (2017).

Van Looy, K., Bouma, J., Herbst, M., Koestel, J., Minasny, B., Mishra, U., Montzka, C., Nemes, A., Pachepsky, Y., Padarian, J., Schaap, M., Tóth, B., Verhoef, A., Vanderborght, J., van der Ploeg, M., Weihermüller, L., Zacharias, S., Zhang, Y. and Vereecken, H. (2017) Pedotransfer functions in Earth system science: challenges and perspectives. Reviews of Geophysics, 55 (4). pp. 1199-1256. ISSN 1944-9208 doi: https://doi.org/10.1002/2017rg000581

Line 152-153: Porosity and bulk density are directly related via a standard soil physical relationship: Porosity = 1.0-BD/density of solids. I am not sure I would call this a PTF 'scheme', although I can see why you want to use this word for consistency.

Line 154: I do not understand what is meant here: "However, with the existence of SOC, soil porosity and water retention capability both tend to increase".

What is both referring to? And what exactly is increasing? If porosity goes up, BD should go down, they can't both increase at the same time? Or did you mean the performance of the schemes increase? But the ones mentioned so far do not use SOC etc.? Also, what is meant with 'existence'? You mean 'availability'. Do we need the word 'capability'? Could we just say 'data'?

Line 156: You menton the 'SocVg scheme'. I presume this refers to Van Genuchten, but not enough information is given here. This needs to be made clearer.

Can you not provide the reader with the exact equations in an appendix? You do in fact, but you are not referring to it here, as far as I can see.

Line 157: Replace 'depict' by 'calculate'.

Line 160: Start a new paragraph when introducing the SHPs. The previous sentence needs to sit with the porosity paragraph.

Lines 160-161: You need to refer the reader to Section A2 in your appendix for the exact CH and VG equations.

Line 161: Replace "suction pressure" with "matric pressure" as suction is always deemed positive.

Line 164-165: You say: "Furthermore, the selected PTFs (see Appendix, Table A1) combined with the optimal porosity scheme were used to estimate the SWRCs-CH and SWRCs-VG."

I am not sure why you choose to make a separation between porosity and the other parameters in your water retention curves. The Cosby et al univariate equation for porosity is also a PTF function?!

Lines 165-166: You say "These two estimations (e.g. measured vs. PTFs) were compared, and suitable PTFs with the lowest bias were selected.". You mean: " the estimates of water retention curves (using Eqs. presented in Section A2) and measured WRCs were compared and the PTFs that yield parameters that resulted in the lowest bias between measured and modelled WRC were selected? Could that mean that a mix of PTFs was used for calculation of WRC? E.g. Cosby for pososity, Brakensiek for the b-parameter etc.? I find this very hard to follow.

Line 169-170:" To estimate the Ks of a mixture containing gravels, Peck and Watson (1979) used a heat-flow analogy correlating with the Ks of fine minerals and the volumetric gravel fraction".

Please rephrase, I find this hard to follow. The fine minerals are in between the gravel. What exactly was correlated with what?

Also, where is the F coming from in the VGF scheme? I don't see the logic here.

Line 178: You mean 'standard' PTFs or 'PTFs from the literature'?

Line 179: You say: "Several (semi-) empirical models have been developed to estimate the STP". There is not one single STP. There are 3 STPs: thermal conductivity, heat capacity and thermal diffusivity. In this case you meant lambda. Remind the reader again what this symbol stands for.

Line 185: Say: "Farouki (1985) proposed an alternative method and..
Line 186: say: "soil minerals".
Line 188/190: Did you mean 'factor' instead of 'feature'?
Line 193-194: You say: "For each $\lambda$ scheme, a comparison was made using parameters (i.e. the $\lambda$ of minerals) with and without gravel and SOC considerations". This is a bit of a throw-away remark, but later on in your figures and discussion this consideration of gravel and SOC becomes quite important. How did you do this? You need to refer explicitly to Eqs. 21, 22 in your Appendix.

Line 195-196: You need to help the reader by referring to these separate A sections throughout the parameterization section. I found this quite hard to follow without the equations present.

Line 202-203: You say: "across the three climate zones over the TP. In the arid zone.." How do these zones fit with Naqu, Maqu etc.? Please refer consistently to the different locations. Also, Figure 1a does not show rainfall gradients or SM, so it is not obvious where the arid zone etc is, unless one knows the TP well.

Line 216: Is usually sub-humid used instead of semi-humid?

Line 229: A BD of 2.1 seems very high, this must have been nearly pure gravel? Yes, yiu mention that in line 215.

Line 234: 'laying'? You mean 'layering'? 'stratification', 'variation'?

Line 239: the magnitude of mean Ks was **of the order of** 10-5 (m/s).
Line 240: ..exhibiting the **lowest values** of porosity'..
Line 241: '.. exhibited a variation of one order of magnitude..'
Line 241: Differed by 1 order of magnitude?
The English in this section really needs tidying up. I think you have got your orders of magnitude confused.

Line 245: "the negative correlation between saturated hydraulic conductivity and soil organic carbon in soil". Are you sure this is the case? More SOC means a higher porosity so generally higher Ks, unless hydrophobicity plays a role, but then you need to state that.

Line 248: ... say: "Ks reached a minimum".

Line 251-252: tend to change or decline with what? With depth? Be precise.

Line 251: "porosities for layers with a GFF...." See also line 255/257

Line 264: Replace "the heterogeneity issue may still exist", by "heterogeneity may have had an effect on the values of soil properties and parameters throughout our sampling procedures, as with any soil field experimentation"

Line 268 and Figure 7: You refer to these and the TC plots as 'profiles', but in my opinion that is incorrect. A profile is a (plot of) a soil property with depth, but here you are showing the dependencies of Ch and lambda on SM, for different depths. That is a different thing..

Also: at times the Ch goes down even if SM has increased. Why is this? According to theory it increases linearly? Also the shape of the curve is more asymptotic than linear?

Line 270: You say: " as soils dry out" but how do you know that it is not caused by the different Ch values of sand and silt/clay?
Line 271: "Figures 7d&e&f show how the relationship..."

Line 273: You say: "The mean λ ranged from 1.8-0.2 (W m-1 K-1) as the soils dried out", but you mean to say "The mean λ ranged from 1.8 at saturation to 0.2 (W m-1 K-1) as the soil reached air-dry state".

Line 279: .. "had lower lambda values" (not comprised)

Line 280 and below: You need to make it clear that you used in-situ values of sand content and BD to calculate porosity. Also, to me it seems strange to have a scheme based on BD to calculate porosity as they are so tightly related. If you have a correct estimate of particle density should they not give the same answer?

Line 291-295: Gravel is a solid thing, it does not have a porosity. Do you mean a soil packed with gravel 'particles' only?

Line 305: "Taking all these results together, it indicates that the BD scheme delivered the greatest predictive accuracy for estimating porosity in profiles across the three climate zones." This comes at no surprise... (see my previous comment).

Line 309: remove 'and'.

Line 310: SWRC do not 'layer', You mean 'showed distinct variation with depth'.

Line 312 end: add: 'compared to the measured WRCs'

Figure 11: you talk in the legend about Cosby et al. (1984) 1 & 2?? Are these the uni- and multi-variate estimates?

Line 314: Here it is ok to talk about suctions as you use positive values.

Line 316: which Table?

Line 317: PFTs cannot predict SM. You need to do a better job at describing exactly what was done and how the parameter estimates were obtained (in the Materials and Methods too).

Line 353-354: You say "In particular, the PTFs given by Cosby et al. (1984) (1&2) predicted Ks well for the CH model". How is the CH model relevant? I thought you were comparing against measurements? Same goes for the VG model in line 354.

Line 360: "indicating that the estimated Ks for the VG model is less affected by gravels". Again, I thought you were simply comparing Ks measured with Ks modelled (with BC & VG, with different PFTs).

Line 366: I think this section needs a more informative title to bring out the fact you are comparing measured data with theory?

Line 367-369:... bias.. compared to the measurements. You need to make that explicit?

Line 395: Basic soil properties

Line 415-416: But now these WRC equations (or ratherthe PTFs to find their parameters) are used with the independent (SoilGrids etc) datasets, right? You need to make that very clear..

Line 436-441: Make it very clear that it is only texture, BD and SOC that you use from these datasets to derive Ks via the PTFs.

Line 445: You are suddenly talking about a LSM? Also soil hydraulic diffusivity is rarely directly used in LSMs? It is directly related to the SWRC and the K curve, so once you have those two one would not need the diffusivity curve.

Line 450-451: "In the semi-arid zone, all datasets generated CH-D and CH-K similar to those"...

The datasets themselves don't generate anything? Keep your language precise..?

Line 463: ... "slower heat transport'? Maybe say "reduced soil heat fluxes'?

Line 463-464: "and thereby subsequently the higher soil temperature derived from the land surface model, and vice-versa".

What is meant? The LSM will overestimate compared to soil temperature measurements?

Line 469: Did you mean 0.3 m3 m-3?

Line 491-494: "The Cosby et al. (1984) PTFs proved more applicable for SHP estimation by the Clapp and Hornberger (1978) (CH) model, and the continuous Wösten et al. (1999) PTFs for SHP estimation by the Van Genuchten (1980) - Mualem (1976) (VG) model"

This seems quite obvious to me. In fact, one could wonder whether one should even attempt to use VG PTFs to find CH parameters and vice versa...

Line 494: ..' deployed successfully'

Line 498: "This information provided indicative signs for soil parameterization in LSMs." This sentence tells us very little...

Line 499: ' are also used'

Line 506: '.. in LSMs'. There is not just one/a LSM?

Anne Verhoef (a.verhoef@reading.ac.uk)

---

## Referee Comment (RC2) · Anonymous Referee #2 · 9 Mar 2018

Review of ESSD-2107-122, Soil Properties on the Tibet Plateau

Potentially a very useful and important data set. Substantial effort to collect these samples, accompanied by or followed by skilful well-documented sample processing and analysis. Excellent data access tools on the 4TU landing page including nice kml file, easy-to-use Excel spreadsheet and links to many accessary files. (Other ESSD data providers could learn some best practices from this effort and the 4TU site.) Potentially, eventually a very good candidate for ESSD.

But, weak presentation, crowded with marginally-useful information. So much confusion that we miss the central strong points of the manuscript: new availability of valuable data from a difficult but important region, with large (?) potential impact.

So much material to evaluate, very long and difficult task for a reviewer. If reviewers find the information tedious or unconvincing, readers will get discouraged much more quickly and tend to ignore the paper and data for lack of accessibility.

The authors will need to sort through a long list of comments below, most of which I can summarize as: 'What does this <data, graph, analysis> tell us about the Tibetan Plateau?; How could I use it in my own research / modeling?, and - otherwise - why do I need to see this? I believe the authors could produce a much shorter, much clearer and much more useful paper following a clear concise outline:
• What data did we gather and how;
• How does this data fill gaps, geographically and scientifically;
• How does this data compare to prior studies or to other high elevation regions;
• How do we as the data collection experts recommend to process the data for incorporation into land surface models; and
• What encouragements and cautions do we offer to users?
• Summary - how does this data help us understand soil moisture on the Tibetan Plateau.

I believe the authors could eliminate about 60% to 70% of the present content. Some of the remainder could go into an Appendix. This 'thinning' process would allow the authors to focus on what they have provided new and on the potential impact(s) of that new data.

Many comments follow. I recommend the authors read them for specific issues, but not attempt to respond to and revise text in all cases. Instead, give us a shorter clearer product that better represents your efforts.

Apparently not one mention of snow in the entire manuscript. Likewise for 'elevation' - not mentioned. Can one really describe soil measurements for high elevation soils without mentioning snow or elevation?

Line 31: "huge" impact on global climate - probably not. Southern Ocean has a much larger impact on heat and carbon fluxes. At the moment, loss of Arctic sea ice and of northern hemisphere snow cover (including over the Tibetan Plateau!) probably have a larger impact on northern hemisphere circulation patterns than TP by itself. TP certainly has a substantial local and regional impact, including on the Asian monsoon systems.

Lines 31 to 91, Introduction: The introduction as written contains too many threads. After a brief geographic description, the authors take readers on a wandering tour through: how and why Land Surface Models work or fail; how other research groups have developed and used PTFs; how LSMs use PTFs; how one needs soil hydraulic and thermal properties to understand and predict soil moisture; how a long list of global observational data sets for soil properties include or do not include observations or correct observations from the TP region.

Some of this material we do not need. Some of it belongs near the end, in a section on impact and future use of the TP data presented here. The sentence at lines 53 to 55, about the need for basic observations of soil properties to understand land-atmosphere interactions and thereby to also improve LSMs, seems like the good place to start. Then how LSM need good parameterisation of hydraulic and thermal properties and how those depend on good

observations, how TP presents a challenge to observations, and therefore the importance of these new data products.

I missed seeing mention of some of the 'third pole' aspects: high elevations at low latitudes cause the unusual presence of ice and snow; precipitation and hydrology difficult to predict; importance both for climate (as mentioned) and for water resources (and biodiversity?); with accurate observations of soil moisture at the centre of all this?

We understand that the authors have extensive technical knowledge and good technical English. Although they thank a colleague for assistance with English translation, the manuscript could benefit substantially from improved science communication. Perhaps the journal can recommend someone who could help from technical English into public-friendly English (assistance that many native English speakers also need)?

Line 97: Sample locations. Figure 1 not useful as presented. Upper left, we do not know what the colours represent? Satellite image of what? We get no sense of terrain / elevation, seasonal snow cover, vegetation if any, etc. Plot the FAO Aridity Index? The browse map on 4TU looks useful, use that? Or use the kml image from Google Earth? In panels a, b and c for the three regions, we do not see elevation contours, vegetation, land use, annual precipitation? Can we even trust the Aridity Index in this region? The explanation of the blue dots vs. the purple triangles needs to appear more prominently, to make clear that it applies to all three panels. In this figure legend you could list the previous sampling networks / sites, then you would not need that information in the text?

Line 110: as written the sentence implies that Yang et al. in 2013 collected the Naqu samples. Because you have already associated the CTP-SMTMN network to Yang et al. 2013 (e.g. in lines 51, 52) you do not need to cite that paper again here.

Lines 112 to 116, "155 total stations ….". This quantitative summary and explanation of sampling constraints should go at the start of the field experiment section? You do not say anything about seasonal freezing of these soils. All samples collected at elevations clearly in the freeze-thaw zone? From the kml file we see Naqu at least 4500m, Ngari at 4800m, even Maqu at 3500m? At those elevations, seasonal snow cover and frozen soil? Elevation terms seem completely missing from presentation and discussion. In the very nice pop-up information boxes for each sampling station in the kml file, all elevations show '0'?

Lines 144 to 149 - Because a reader gets the list of external datasets used for intercomparison here, perhaps we do not need the same information in the introduction?

Line 154 - SOC may have stronger influence on water retention than on porosity, but porosity and water retention can not both increase, e.g. they have an inverse relationship (as you already said in line 152 and as you show in Figure 4).

Line 155 - "properties tend to decrease" Which properties. SOC? Again, porosity and bulk density should not both decrease.

Lines 159 to 161 - Because we get a description of the SWRC schemes - and their associated authors - here, we do not need the same information in the introduction?

Results and Figures - way too much! The authors have shown us everything. Instead they should show us what matters: unique features of these data, where these data confirm or fill gaps in the global soil datasets, the best or preferred parameterisations but not every possible combination? Instead of every possible parameter, what makes these data useful? Figure 2 and the Appendix can guide us through your processes. Show us your best or surprising outcomes! What strengths do these data offer to other users? What weaknesses? What further research - by your group or by others - now becomes possible? Give us the most interesting or useful examples but not every example. Eliminate many Figures? Some tables and figures could go in the Appendix, with the relevant equations. Focus on what new and what valuable these data provide to the research community!

Line 200 and Figure 3, soil texture: Here we need additional information to understand statistical reliability of these soil descriptions.  If the bars in Figures 3a,b and c represent the minimum and maximum values for all sample locations (usually 8 or fewer?) in that region at those depths, then the authors really can not say that sand fraction at 5 or 10 cm exceeds sand fraction at 20 or 40 cm because the range of values appears very wide at all depths.  Give us the sample numbers for each location (and perhaps for each depth if those vary)?  In the very clean and helpful data spreadsheet, I find only the same summary data: mean, min and max.  No indication of sample number or statistical distribution (e.g. std error).  To this reviewer, sand fraction at Ngari not statistically different from sand fraction at Nadu and no reliable vertical pattern at either location?  Figure legend says minimum and maximum values in the "profile".  The authors must mean in all profiles (note the plural) from that region because they have given us no hints of replicate samples at any depth of any profile.  Only GGF varies between arid and semi-arid?  Otherwise the soil texture at these two sites appear the same.  Maqu statistically different to the other two sites but with no significant vertical variation except perhaps for SOC?  Likewise for the fine and gravel diameters?  We do not get information needed to agree or disagree with statements like this one: "the gravel content increased the deeper the layer." at Nqari (line 208).  This reader does not find these plots useful or surprising.  Do the authors have something special to show us about TP soil texture?  Or do these profiles look as expected?  Either way, we don't need to see this?  And we don't need more than a sentence or two of description?  Could the authors make the raw data available?  E.g. instead of listing only mean, min and max in the spreadsheet, could we get all the station data for each region so that we can plot our own statistical distributions and vertical profiles?

Line 223 and Figure 4, bulk density and porosity.  Same statistical problem: seeing maximum and minimum tells us nothing about distribution.  At least we can see different patterns for different regions but again, do the authors want to call our attention to anything special about TP soils?  If not, why do we need to see this figure or read this discussion?  We have no statistical basis to accept this statement (line 233): "The stratification of BD and porosity in the profile might be induced by SOC laying as Fig. 3c reveals." because we have no confidence in the so-called SOC layering - except perhaps at 5 cm depth - in Figure 3c.

Line 238 and Figure 5, hydraulic conductivity.  Same statistical problem.   With a log scale for the abscissa, this reader does not see any reliable vertical pattern or station difference between Ngari and Nadu.  Magu might differ from the other two and might show one vertical outlying value.  Anything special about TP here?  The mean and st dev (from the supplement, mentioned in line 248) might prove more relevant here and then put this figure in an Appendix or supplement?

Line 250 and Figure 6, GGF with porosity and hydraulic conductivity.  Nothing useful here at all?  Anything that would distinguish these as TP soils?  At least we can visually get some idea of number of samples?  For 5 cm at these two locations, 5 or 6 samples in one case and perhaps only 3 for the other case?  This reader finds no validity nor any information useful to TP in either text or figure.  No significant slopes or significant R-square values for any depth at either location?  Suggest that they really do not have enough reliable data collected across a wide-enough range of station locations and soil types to make this analysis useful?  The sentence in lines 264, 265 represents everything we need to know here?

Line 267 and Figure 7, heat capacity and thermal conductivity.  If "no distinct stratification" then we really don't need to see 7a, b and c?  For 7d, e and f, these statements assume that we have accepted vertical stratification of soil texture above. If we have not, if the authors have not helped us by identifying statistically reliable patterns, then these figures and the associated paragraph have no utility.  Again, anything special about TP or do these look like all other soils?

Line 280 and Figure 8, porosity measured and predicted.  No need for all these correlation figures, you could do it all with simple correlation coefficient values.  Or a simple statement: BD provided the best fit across a range of depths and porosities, as in lines 305, 306, then refer reader to Appendix.  We do not need to see all these mis-fit figures.  How do these data compare to any other high-elevation soils?  Anything special or need special attention about the TP soils?

Line 308, SWRC comparisons.  Figure 9: nothing useful. CH and VG not distinguishable at any station.  Maqu different to Ngari and Naqu, but we already knew that.  What, if anything, proves

specifically relevant to TP? I doubt the utility of this figure but if you keep it, it should go in Appendix or supplement. Figure 10: (why back to colour?) Again, no useful information by station, depth or parameterisation. If nothing unusual or relevant to TP, omit this figure. Figure 11: possibly some useful information here but one could do it in a small table rather than this mostly-empty (visually and intellectually) table. 90% of bias values identical regardless of station, approach or author. Overall we really don't need more than the concluding sentence (lines 346 to 348) for this entire section. How does this conclusion compare to other soils or other high-elevation soils? Anything special about TP here?

Line 349 and Figure 12, hydraulic conductivity: Colour, but nothing to help a reader extract useful information from this figure. Small biases for many parameterisations, but relevance? If you give us the summary (lines 362 to 364) we don't need to see this figure? Should we learn anything special about TP soils from this discussion?

Line 366 and Figures 13, 14, heat capacity and thermal conductivities. By now this reviewer feels like a broken record, repeating endlessly the same messages. I find nothing in this text or figures that shows me anything useful about TP soils or about the relevance of various parameterisation schemes to TP soils. We do not need to see any of this?

Line 395 and Figures 15 and 16, comparison of basic soil properties: This information should lie at the heart of this paper, to help users understand the impacts of this new data. Figure 15 potentially useful but it has too much white (empty) space. Because many potential users will skip text but look at this figure, the authors should define the data source acronyms in the Figure legend. In Figure 16, these observations for soil porosity in all three regions appear as clear outliers, at many if not all depths, to all the existing data sources. Why? What impact? What do the authors want us to know about the strengths or weaknesses of their data compared to the existing products? Do the authors have something special to say about TP soils? Missing answers to this question: how do these data help fill scientific or geographic gaps?

Line 414 and Figures 17, 18, 19, soil water retention curves: Good initial point that if the existing soil descriptions have uniform (but, inaccurate?) soil porosity profiles they will also necessarily have wrong water retention curves. Figure 17 emphasises this point, e.g. not possible to extract useful information from that figure? Suggest the authors omit Fig 17. Figure 18 potentially useful - first time we might actually see an impact of the TP observations - but visually not helpful. Most schemes, especially VG in the lower panel, look identical regardless of observation sources. If all the prior observations tend to converge on wrong values, then one would expect this uniformity? Perhaps lump all the prior obs with an uncertainty term and then show the impact of the current data? This statement (line 432) "With the Tibet-Obs dataset as input to the applicable PTFs good FC and PWP were generated." seems to defeat this entire effort, because measurements presented here do not converge with so-called Tibet-Obs and Tibet-Obs existed prior to this extensive data gathering effort! How do the present observations impact PTFs? We never get a clear statement? Figure 19, same concerns as Fig 18. If all the prior obs converge to a uniform but wrong data value, then all the hydraulic conductivity outcomes will necessarily look basically the same? Average and lump them all, then show the new data for comparison? Chance for a strong impact here but lost because of the vague presentation.

Lines 442 to 477, figures 20 to 22. Possibly these figures actually contain useful information showing impact of these observations? Typical author approach - show us all curves in every possible combination - defeats what might have appeared as useful information. Too many plots look indistinguishable: panel a and c in Figure 20, all panels in Figure 21, all panels except 'a-right' in Figure 22. (Figure 22 looks almost useless?). But the authors could help our eyes find useful information: where the SHP and STP factors differ substantially (higher or lower) from the observations? Reading the text, I found only a narrative version of the indistinct graphics: sometimes overestimated, sometimes underestimated, everything treated the same, no clear outcomes. if no clear outcomes, leave this out entirely? If distinct and useful outcomes, highlight those?

Personally I found this disappointing. At the end of all this work, at the conclusion of careful analysis, we basically get more curves and scatter plots that tell us almost nothing? If true, then the authors should simply present the data with its strengths and uncertainties and forego all of

this analysis, leave all analysis to others or to a subsequent research paper?  If, instead, the authors have tangible results and unique features to point out, then they need to take a much clearer shorter route to explicit and compelling outcomes.  Their approach of plotting all possible variables and schemes in all possible combinations simply doesn't work.

Line 478, Data availability: Technically very good as already mentioned.  Spreadsheet contains the averaged values rather than the raw data, which prevents users from doing their own statistical analysis.

Line 483, Conclusions: Paragraph one, that the soil properties vary across climate zones and within profile seems not surprising and perhaps - because of the absence of necessary statistics - not valid.  In any case, why would one expect anything different for any soils anywhere on the planet.  What, if anything, do these data show us about TP specifically?  Paragraph two, that some schemes work better than others seems also not surprising.  We could have learned this from any series of soil samples taken anywhere in the world?  Did we learn anything peculiar about high elevation soils generally or about TP soils in particular?  Paragraph 3 about how these data impact (or improve?) estimations SHP and STP and about relevance to remote sensing of SM seems potentially useful, but we miss any explanation or emphasis about how these data have filled scientific or geographical gaps.  We get only tentative recommendations applicable to each of  the climate zones for the TP, but no assessment of if and how these data impact our basic understanding of those proposed climate zones?  For what part of the year can satellites actually attempt to measure soil moisture (as opposed to snow cover) at these TP sites?  Paragraph four seems like a short useful summary but it implies that the authors only evaluated the applicable schemes.  It omits the essential fact that - apparently for the first time - we actually had observational data on which to base the evaluations of each scheme.

---

## Author Comment (AC1) · 3 May 2018

Dear Prof. Anne Verhoef,

We sincerely thank you for your quality review on our manuscript. The constructive comments and suggestions are very useful to improve the quality of our paper. We carefully considered each point and revised the manuscript accordingly. Please find our responses to the comments in the supplement and the related changes in the modified manuscript.

Please also note the supplement to this comment:

[Figure]

https://www.earth-syst-sci-data-discuss.net/essd-2017-122/essd-2017-122-AC1-supplement.zip

---

## Author Comment (AC2) · 3 May 2018

Dear Reviewer 2,

We are grateful for your constructive comments and suggestions. The comments are considerably useful to improve the quality of our paper. We've adjusted the structure and carefully revised the manuscript accordingly. Please find below your comments with our point-to-point responses as well as the corresponding changes in the modified manuscript.

Please also note the supplement to this comment:

https://www.earth-syst-sci-data-discuss.net/essd-2017-122/essd-2017-122-AC2-supplement.zip

---

## Author Comment (AC3) · 8 May 2018

Dear Reviewers,

We sincerely thank you for your quality review on our manuscript. Please find the updated modified manuscript-tracked, manuscript-cleaned and supplement in the attachment.

Please also note the supplement to this comment:
https://www.earth-syst-sci-data-discuss.net/essd-2017-122/essd-2017-122-AC3-supplement.zip